# Targeted perturbations reveal brain-like local coding axes in robustified, but not standard, ANN-based brain models

## Abstract

Artificial neural networks (ANNs) have become the de facto standard for modeling the human visual system, primarily due to their success in predicting neural responses. However, with many models now achieving similar predictive accuracy, we need a stronger criterion. Here, we use small-scale adversarial probes to characterize the local representational geometry of many highly predictive ANN-based brain models. We report four key findings. First, we show that most contemporary ANN-based brain models are unexpectedly fragile. Despite high prediction scores, their response predictions are highly sensitive to small, imperceptible perturbations, revealing unreliable local coding directions. Second, we demonstrate that a model's sensitivity to adversarial probes can better discriminate between candidate neural encoding models than prediction accuracy alone. Third, we find that standard models rely on distinct local coding directions that do not transfer across model architectures. Finally, we show that adversarial probes from robustified models produce generalizable and semantically meaningful changes, suggesting that they capture the local coding dimensions of the visual system. Together, our work shows that local representational geometry provides a stronger criterion for brain model evaluation. We also provide empirical grounds for favoring robust models, whose more stable coding axes not only align better with neural selectivity but also generate concrete, testable predictions for future experiments.

## 1 Introduction

For over a decade, NeuroAI has celebrated artificial neural networks (ANNs) for how well they predict brain responses (Yamins et al., 2014; Kriegeskorte, 2015; Storrs et al., 2021; Zhuang et al., 2021; Doerig et al., 2023). However, the field now faces a new challenge: a diverse array of ANN models predict data equally well, making it nearly impossible to distinguish between them using accuracy alone (Schrimpf et al., 2018; Conwell et al., 2023; Linsley et al., 2023; Ratan Murty et al., 2021). This convergence between ANN models compels us to ask a new set of questions. If multiple models predict the brain equally well, are they truly meaningful and equivalent representations of the brain? To find out, we need more precise tests. Here, we ask a very simple question: how much does it take to alter a model's predictions? We designed small-scale adversarial probes to test this question and find that even our best ANN-based brain models are remarkably fragile, though to different degrees (Sections 1 and 2). We then leverage this observation to characterize each model's local coding directions (Section 3) and to generate testable predictions for future human and animal experiments (Section 4). Our systematic analyses of local representational geometry of brain models shows that robustified models, unlike standard networks, better capture the stable local coding axes of the brain. These models set the stage for the next tests, experiments that will directly probe and manipulate neural representations.

A major source of ambiguity in why different ANN-based models predict neural responses equally well lies in the methods we use to map model features onto brain data. In practice, we do not directly compare model features to neurons/voxels. Instead, they are *encoding* models that learn a linear readout (eg. ridge regression) between units in a specific model layer and the brain, selecting and reweighting model features in the process (Kay et al., 2008; Mitchell et al., 2008; Naselaris et al., 2011; Yamins et al., 2014). This mapping is assumed to harmonize neural network representations

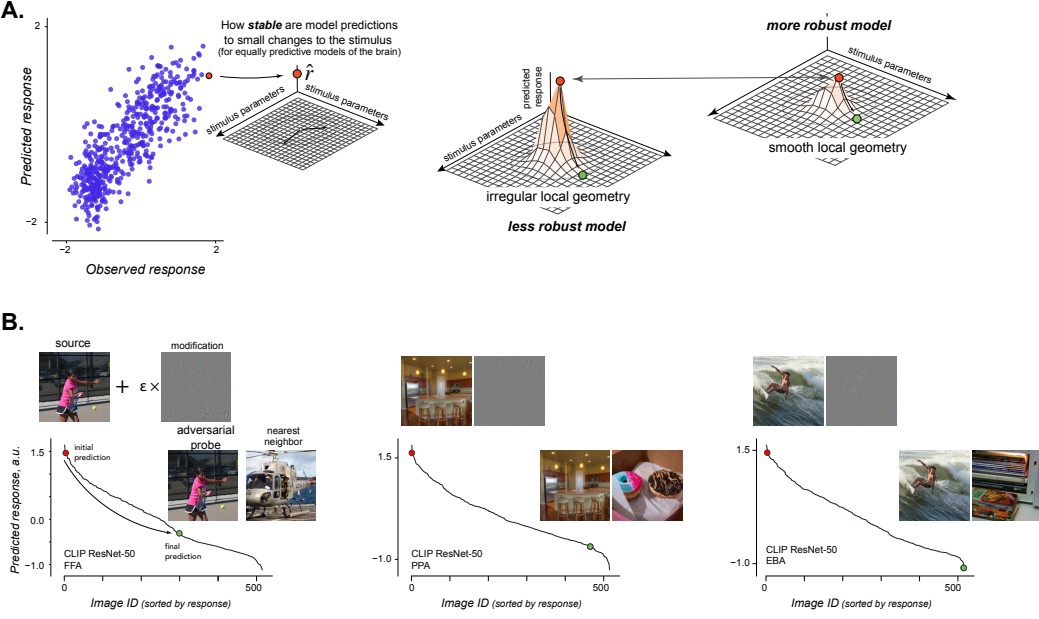

Figure 1: **Adversarial sensitivity reveals local representational geometry.**
**A:** Scatterplot showing predicted versus observed responses for a voxel in the FFA using CLIP ResNet-50. Each dot corresponds to a held-out stimulus. Insets illustrate the central question: how stable are model predictions to small input changes? On the right, schematic examples show two possibilities. In a less robust model (irregular local geometry), a small perturbation can cause a large shift in predicted response. In a more robust model (smooth local geometry), the same perturbation results in only minor changes. **B:** Examples of adversarial probes for CLIP ResNet-50 in three brain regions (FFA, PPA, EBA respectively). In each case, the y-axis shows predicted response and the x-axis shows held-out images ranked by prediction. A source image (red dot) initially predicted to elicit a strong response shifts dramatically (green dot) after an imperceptible perturbation. Insets show the source, perturbation, modified image, and nearest-neighbor control.

by projecting them onto a shared, brain-aligned response subspace. Less explored, however, is the degree to which the resulting mapped model (or brain model) inherits the properties and vulnerabilities of the underlying ANN. One possibility is that the brain-alignment procedure downweights the idiosyncratic ANN-specific representations and emphasizes the brain-relevant ones. By this account, all equally predictive brain models should be similar. Another possibility is that the linear readout simply amplifies the features most predictive in the dataset, even if they are fragile and unrelated to the brain. By this account, each model, even if equally predictive of the brain, is distinct. In either case, the resulting models are treated as if they *are* the brain and embody the neural coding axes – an assumption that underlies much of the current NeuroAI enterprise.

The way to decide between these possibilities is to probe the local geometry of the resulting brain models around an image. To do this, we used adversarial probes: small-scale, often imperceptible, identity-preserving tweaks to the stimulus directly optimized to change the predicted response (Szegedy et al., 2014; Goodfellow et al., 2015; Kurakin et al., 2017; Croce and Hein, 2020; Su et al., 2019; Moosavi-Dezfooli et al., 2017; Xiao et al., 2023). Adversarial probes, when used in context of brain models, can reveal how steep the response landscape is around a given image. If tiny nudges produce large shifts in prediction, the local geometry is sharp and irregular, and the model is unstable. If predictions barely move, the geometry is smooth and the model is robust, likely closer to the brain. Figure 1A illustrates this idea. The horizontal axes represent stimulus parameters and the vertical axis is the model's predicted response ($\hat{r}$). The red dot marks the unaltered image and concentric rings indicate equal-sized perturbation budgets ($\epsilon$) but of increasing magnitudes. In the less robust case (middle), even a small change in the stimulus would result in an unusually large change in $\hat{r}$, consistent with complex local geometry. In the more robust case (right), the same-size change

produces only a minimal shift in response, consistent with smooth local geometry (more robust). If models with equal predictive accuracy show similar effects of perturbations, this would suggest that the brain models have shared local coding axes and a common brain-aligned geometry. If they respond differently and attacks do not transfer, it would indicate that predictivity masks important differences: the models rely on distinct, model-specific axes, and their local geometries diverge from one another and from the brain. To our knowledge, these aspects have not been systematically tested.

**Related Work:** While ANN-based encoding models are now central to NeuroAI, there has been no systematic study of how adversarial perturbations affect brain models themselves. In particular, the local coding directions of ANN-based brain models and the fine-scale geometry of their representations remain essentially unknown. By contrast, in machine learning, adversarial perturbations have been extensively used to reveal discrepancies between ANNs and human perception (Elsayed et al., 2018; Zhou and Firestone, 2019) and to develop robust training methods (robustified models) (Madry et al., 2018; Tramèr et al., 2018). A few studies have brought adversarial methods into neuroscience, but with a different emphasis: some have used adversarial images to modulate behavioral responses in humans (Gaziv et al., 2023), others have introduced statistical eigen-distortion tests to compare pairs of models (Feather et al., 2024; Berardino et al., 2017), and some studies leverage robustified models trained with neural data to design perceptible stimuli to drive brain responses (Guo et al., 2022; Gaziv et al., 2025). None of these approaches address the stability of our commonplace encoding models that dominate current practice. Our study is, to our knowledge, the first to systematically characterize adversarial sensitivity and perturbation subspaces in ANN-based brain models, providing a new lens on brain-model alignment.

We make four core contributions: 1) We show that contemporary ANN-based encoding models of the brain, though highly predictive of brain responses, are unexpectedly fragile. Small, imperceptible adversarial probes can substantially disrupt model predictions. 2) We demonstrate that a model's sensitivity to adversarial probes provides a stronger criterion for distinguishing between equally predictive models of the brain than predictivity alone. 3) We show that adversarial probes are highly specific and often fail to transfer across models. Different ANN-based brain models occupy largely distinct perturbation subspaces despite comparable prediction accuracy. 4) We identify perturbation probes that consistently affect multiple encoding models, which we speculate might reflect latent coding dimensions of the human visual system. Taken together, our findings establish local representational geometry as a critical dimension of model evaluation, highlight robustified models as better aligned with local coding directions, and position adversarial probes as a principled tool for understanding small-scale representations and generating causal predictions about the brain.

## 2 METHODS

**Voxelwise encoding Models**: An ANN-based encoding model has two components: features, or embeddings, from a specific layer of the artificial neural network (the representational basis) and a trainable readout (mapping) function. The readout is typically done through regularized linear regression, which projects the features into the response subspace of neural activity. Formally, each training image is passed through a pre-trained encoder $f$ yielding a latent feature tensor $z_l \in \mathbb{R}^{C_l \times H_l \times W_l}$. These features are then passed through a mapping function $g : \mathbb{R}^{C_l \times H_l \times W_l} \to \mathbb{R}^m$, where $m$ is the dimensionality of the neural data being predicted (e.g., number of voxels). The encoder $f$ is kept fixed and only the readout $g$ is trained. In our study, we flatten $z_l$ into a vector and use ridge regression to construct the readout mapping $g$ with a regularization coefficient chosen through nested cross-validation. We considered 14 pre-trained artificial neural networks previously validated against brain data. In addition, to investigate whether increasing robustness improves the prediction accuracy of the encoding models, we also used publicly available models that were robustified through adversarial training (Engstrom et al., 2019; Ilyas et al., 2019). These models share the same architecture (ResNet-50) and learning rule but differ in the degree to which they are trained adversarially. Further details on our encoding models can be found in Appendix A.1.

**fMRI Dataset:** We used publicly available 7T fMRI data from the Natural Scenes Dataset (NSD) (Allen et al., 2022) for all analyses in this study. We focused on the responses to 1000 shared stimuli obtained from fMRI scans of four subjects in category-selective brain regions. Each subject viewed these images three times over multiple experimental sessions. All analyses were conducted using

version 3 of the dataset (*betas_fithrf_GLMdenoise_RR*), obtained directly from the NSD website. In this work, we focused on the category-selective areas: fusiform face area (FFA) (Kanwisher et al., 1997), extrastriate body area (EBA) (Downing et al., 2001), and the parahippocampal place area (PPA) (Epstein and Kanwisher, 1998). To ensure the inclusion of only the most category-selective voxels, we applied a stringent threshold of $tval > 7$ for all analyses. Models were trained to predict the voxel and trial-averaged responses across subjects, standard in the field.

**Adversarial attack design and evaluation metrics** An adversarial attack seeks to find a small modification to an image $\delta$, bounded by a "perturbation budget" $\epsilon$, predicted to drastically alter the output of a model. A successful attack would significantly (and unrealistically) change the predicted response of the encoding model. We quantified the adversarial sensitivity $s_i$ for a given voxel as the absolute value of the change in response, comparable to the method used in Guo et al. (2022). Specifically, we define a sensitivity measurement $s_i$ for the $i$-th voxel as:

$$s_i = \max_{||\delta||_p \leq \epsilon} |r - \hat{r}|,$$

where $r = g(f(x))$ and $\hat{r} = g(f(x + \delta))$.

There are two things to note about this metric. First, since $s_i$ is a measure of model *sensitivity*, high values on this metric would indicate lower adversarial robustness. The second is that since the metric does not have an upper bound, the results must not be interpreted across regions. Importantly, we did not find that normalizing voxel responses (by z-scoring or min-max) had any significant effect on our results. We ran two adversarial attacks per image (one to minimize and one to maximize the predicted response), and we selected the version resulting in the larger $s_i$ for analyses. In total, over all models, regions, subjects, voxels, attack directions, and attack types, we perform nearly two million adversarial attacks.

We report our results regarding sensitivity to $l_2$-bounded attacks, although all results hold for $l_\infty$-bounded attacks as well. To find our adversarial attacks, we use an iterative gradient descent method (for example, for $\epsilon = 5$, we take five equally spaced steps in the $l_2$-ball). Further details on the adversarial attacks, along with the results for $l_\infty$-bounded attacks, can be found in Appendix A.2 and A.3.

## 3 RESULTS

All experiments were performed on human fMRI data from the Natural Scenes Dataset (NSD), focusing on high-level visual regions with well-established category selectivity: face (FFA), body (EBA), and scene (PPA). We chose these regions because their response profiles are well understood, providing a strong foundation for interpreting adversarial probes. For example, the FFA responds strongly to faces and weakly to scenes. This predictable selectivity makes them ideal test cases for asking whether adversarial noise disrupts established patterns and whether adversarial probes shift responses along meaningful neural tuning axes or push them into idiosyncratic, uninterpretable directions. We also restricted most of our analyses to very small image perturbations imperceptible to humans (especially for claims supporting parts 1 and 2). This is important because the effects of targeted noise patterns on brain voxel responses is unknown. We confirmed that $\epsilon = 5$ was below the perceptual threshold for noise detection, based on pilot data from a simple image discrimination psychophysics experiment.

**Section 1: ANN-based encoding models are highly susceptible to small-scale adversarial noise**

We first sought to confirm that diverse ANN-based encoding models predict neural responses with similarly high accuracy. As in prior work, we identified the most predictive layer for each subject and ANN model and tested model performance on held-out data (see Methods). We observed that ANN-based models were highly accurate (Figure 2A) and the differences in prediction scores between model architectures was minimal (normalized variance across models = 0.001). This analysis replicates prior results and highlights a key challenge in the field: predictive accuracy alone does not meaningfully distinguish between candidate ANN-based models (Canatar et al., 2023; Tuckute et al., 2022; Conwell et al., 2023; Schrimpf et al., 2020; Ratan Murty et al., 2021).

If all models predict responses equally well, should they be considered the same? We know that responses in the brain are reliable across repeated presentations and robust to small, irrelevant changes

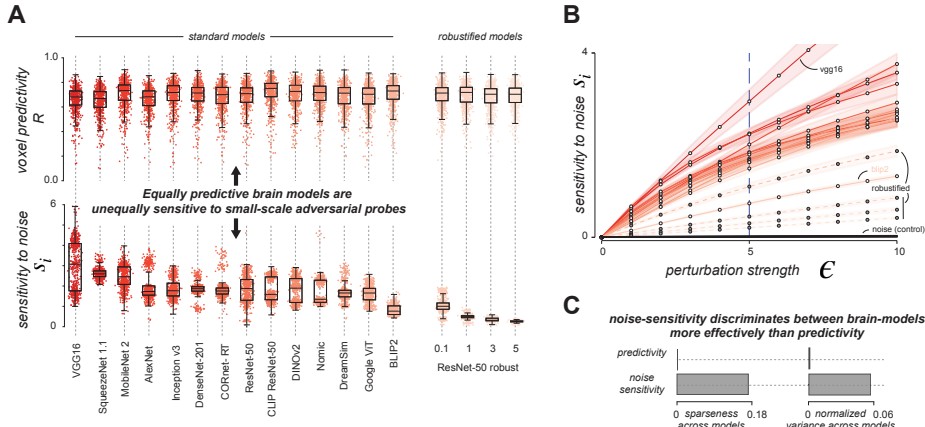

Figure 2: **Adversarial sensitivity provides a sharper test of brain models than predictivity.**
**A:** Top: Boxplots showing the predictive accuracy of candidate encoding models (x-axis) against brain responses (y-axis). Boxes indicate the median prediction accuracy with error bars for voxel-level standard error; dots correspond to individual voxels. Bottom: Boxplots showing adversarial sensitivity (y-axis) for the same models, measured at a perturbation budget of $\epsilon = 5$. **B:** Adversarial sensitivity functions. The x-axis indicates perturbation strength ($\epsilon$) and the y-axis shows model sensitivity. Lines represent different models. Standard models (solid) exhibit steep increases in sensitivity even at very small perturbations (e.g., VGG16), whereas robustified models (dashed) are more stable, requiring larger perturbations to shift predictions. The black control line shows randomized noise, which has minimal effect **C:** Discriminability of models. Left: sparseness across models. Right: normalized variance. Both measures show that noise sensitivity separates models more effectively than predictivity, demonstrating that adversarial sensitivity is a stronger criterion for distinguishing candidate brain models.

in the input. How stable are the predictions from these highly accurate ANN-based brain models? To address this question, we designed adversarial probes. We engineered imperceptible changes to images that drastically shift the response of the brain models and compared the observed response against shuffled noise of the same magnitude and statistical properties (negative control). We reasoned that if perturbations were small and imperceptible to humans, then models designed to approximate brain responses should not change their predictions either. We first present some exemplars based on CLIP ResNet-50, an ANN architecture widely used in neuroscience (Figure 1B).

In the case of CLIP ResNet-50 for the FFA (a face-selective region), a face image elicited a high predicted response (as expected for this brain region). Adding a barely visible adversarial perturbation, however, drove the prediction to the extreme end of the response spectrum, well outside the expected range. Importantly, the shuffled control noise of the same intensity had little effect to no effect on the model predictions, indicating that the change in response was highly specific. Notably, this suggests that the local coding directions are highly specific (and informative) for each model. It reveals that models with matched neural predictivity encode distinct, structured, and meaningful local directions that standard prediction metrics completely miss. This is not solely a consquence of injecting out-of-distribution noise to the images, as shown by the shuffled control with equal statistics as the adversarial noise. We show the results for all model architectures in Figure 2A. We next asked how these models compared against models trained with adversarial robustness objectives ("*robustified* models" here). These models (Figure 2A, right) showed substantially reduced sensitivity, setting them apart from standard networks that, despite achieving similar prediction accuracies, were far more fragile.

Next, we estimated how strongly each model's response predictions shifted as a function of the perturbation strength (the *adversarial sensitivity function*). This tells us how stable a model is (y-axis) when nudged by increasing amount of targeted noise (perturbation budget, $\epsilon$, x-axis, concentric rings in Figure 1A). Figure 2B shows these sensitivity functions across models. As perturbation

strength increased (x-axis), model sensitivity also increased (as expected). All standard models were highly sensitive even for the smallest perturbations we evaluated ($\epsilon = 1$). Some models like BLIP2 were relatively more robust.

At this point we obviously wondered how standard models compared to *robustified* models. These models were trained explicitly to be robust to increasing amounts of adversarial noise. We found robustified models (dashed lines) to be considerably more stable than standard models without robust training. These results demonstrate that ANN-based brain encoding models (mapped to neural data) inherit the vulnerabilities of contemporary neural network models. Distinct model architectures may predict neural responses with high accuracy, but those predictions themselves are quite fragile and can be easily nudged by targeted imperceptible noise.

**Does robustifying a model trivially explain local coding directions?** We note that adversarial training is specifically designed to reduce sensitivity at the last layer of a neural network (the classifier). Here, we instead investigate the sensitivity at the *neural response level*, regressed from intermediate activations. We focus on the adversarial sensitivity at the neural response level solely in this study, as this behavior remains unknown and is not necessarily implied by sensitivity at the classifier level.

### Section 2: Adversarial sensitivity better discriminates between high-performing encoding models of the brain

Here, we asked whether sensitivity to targeted perturbations separates models better than predictive accuracy alone. We quantified how well each metric distinguishes among models using two complementary measures. First, we computed sparseness across models. Sparseness is scale-invariant metric often used in non-human primate neuroscience which lets us compare predicitivity and adversarial sensitivity on a equal footing. As shown in Figure 2C, sparseness was significantly higher for adversarial sensitivity than for predictivity, indicating that sensitivity provides a greater spread and thus better discriminability across models. We also computed the normalized variance between the two scores to provide a more familiar dispersion measure (and to allay a possible concern that sparseness values may be driven by outliers). The normalized variance was also higher for adversarial sensitivity than predictivity (Figure 2C). These results are consistent and show that adversarial sensitivity better discriminates between candidate ANN-based brain models models than prediction scores alone. More details regarding the sparseness and normalized variance metrics can be found in A.1

### Section 3: ANN-based encoding models have distinct perturbation subspaces

Up until now, we have evaluated models *one at a time*. These results show that adversarial probes are potent and can distinguish among ANN-based models that are otherwise highly predictive of brain responses. In this section, we go further and characterize how these models *relate to one another*. Does an adversarial probe that disrupts one model also affect other models? In other words, do different models share common vulnerable directions, or does each model rely on its own idiosyncratic coding axes? We selected 50 top voxels for each subject and brain region with the highest model signal-to-noise ratio and prediction accuracy and used their average response as the target. This SNR-based selection was independent of the adversarial procedure and ensured that analyses focused on reliable voxels. We attacked each model with a fixed small perturbation budget ($\epsilon = 5$, as before) and a single optimization step. This procedure isolates the first-order sensitivity of each model (its dominant gradient at the clean image). We then tested whether the resulting perturbation probe transferred to other models. This procedure characterizes the local representational geometry of the resulting brain models: if equally predictive models share the same direction of maximal sensitivity in image space, the single-step perturbation should transfer; if transfer is weak, the models rely on distinct local coding axes (see Figure 3A).

Figure 3B shows the transfer matrix for all models, with columns indicating the source model on which the adversarial probe was crafted, and rows indicating the target model to which it was applied. In the upper-left block, corresponding to standard architectures, probes that strongly disrupted the source model had little effect on other models, indicating that these architectures rely on distinct local coding axes. The lower-right block shows results for the robustified models. Here we observe an asymmetry. Adversarial probes from standard models have little effect on robustified models, but adversarial probes from robustified models generalize relatively more effectively, though still not to the same degree as to their own model. If models encoded stimuli along the same perceptually

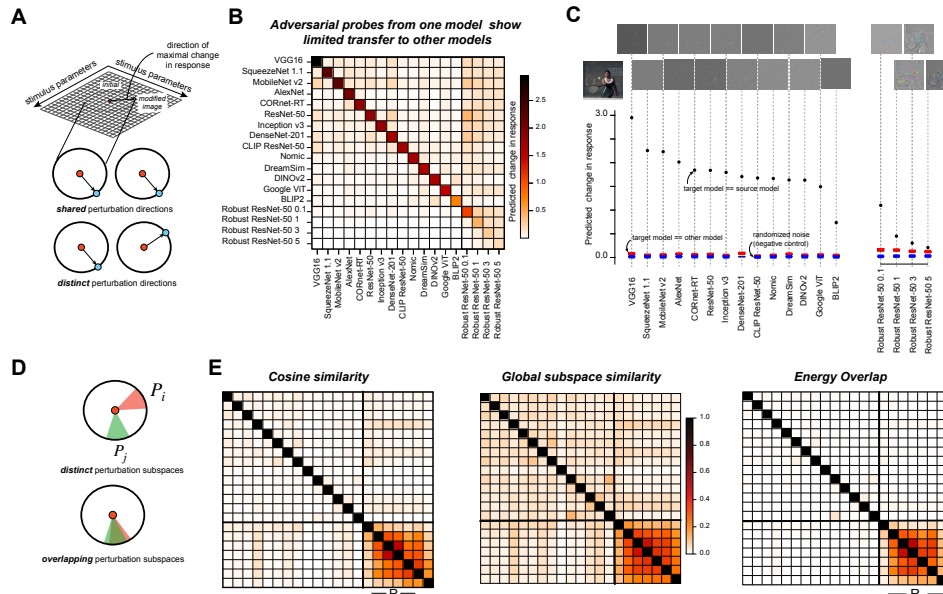

Figure 3: **Adversarial probes reveal model-specific coding axes and distinct perturbation subspaces**
**A:** Schematic contrasting two hypotheses: models may share or differ in their perturbation directions. The question is whether the direction of maximal change for one model also modulates other models. **B:** Transfer matrix of adversarial probes ($\epsilon = 5$). Strong self-effects (diagonal) contrast with weak transfer across models, especially among standard networks. Robustified models show somewhat greater transfer. **C:** Plot illustrating encoding model adversarial transferability. The black dots represent the adversarial sensitivity of a model to attacks sourced on itself. Red boxes indicate the model's transfer strength to other models (the average of that model's row in (B), minus the identity cell). Blue boxes indicate the negative control (model's transfer strength when the attack is randomized noise). **D:** Schematic contrasting two hypotheses about perturbation subspaces: distinct vs overlapping. **E:** Three metrics representing similarity between perturbation subspaces. We include three additional robustified architectures (VGG, MobileNet, and DenseNet) to diversify the set of models compared.

meaningful axes, the same small nudge would move them all. Instead, we find that standard models respond along different axes with little transfer.

Importantly, this does not imply there *does not exist* a universal perturbation which will affect all models. A large body of work has investigated adversarial transferability across models (Xie et al., 2019; Tramèr et al., 2017; Gu et al., 2023). These studies typically use multi-step, high-norm, or perceptibly altered perturbations that are explicitly optimized to force transfer across classifiers. Here, we instead utilize low-norm, first-order probes to study the local coding directions in neural-prediction space.

The single-step transfer test (above) probes only the single, most sensitive local direction. However, brain models may be vulnerable along a multi-dimensional subspace. To investigate this possibility, we extended our analyses to the full *perturbation subspace* of a model: the set of directions in pixel space to which a region's response is locally sensitive. For a given model, the first order sensitivity of a multivoxel response $r \in \mathbb{R}^k$ (with $k = 50$ top voxels) to an input image $x \in \mathbb{R}^p$ is fully described by the Jacobian matrix $J \equiv \frac{\partial r}{\partial x} \in \mathbb{R}^{k \times p}$. The directions in pixel space that produce the largest changes in the multi-voxel response pattern are captured by the right singular vectors of $J$. These vectors form an orthonormal basis for the model $i$'s perturbation subspace $\mathcal{P}_i$. $\mathcal{P}_i$ is the subspace spanned by the top-$k$ singular vectors.

We quantified the geometric alignment between the perturbation subspaces of two models $\mathcal{P}_i$ and $\mathcal{P}_j$ using three different metrics. First, we measured the absolute cosine similarity between the leading directions of sensitivity (the top right singular vectors) from $\mathcal{P}_i$ and $\mathcal{P}_j$, $|v_{i,1}^\top v_{j,1}|$. Second, we

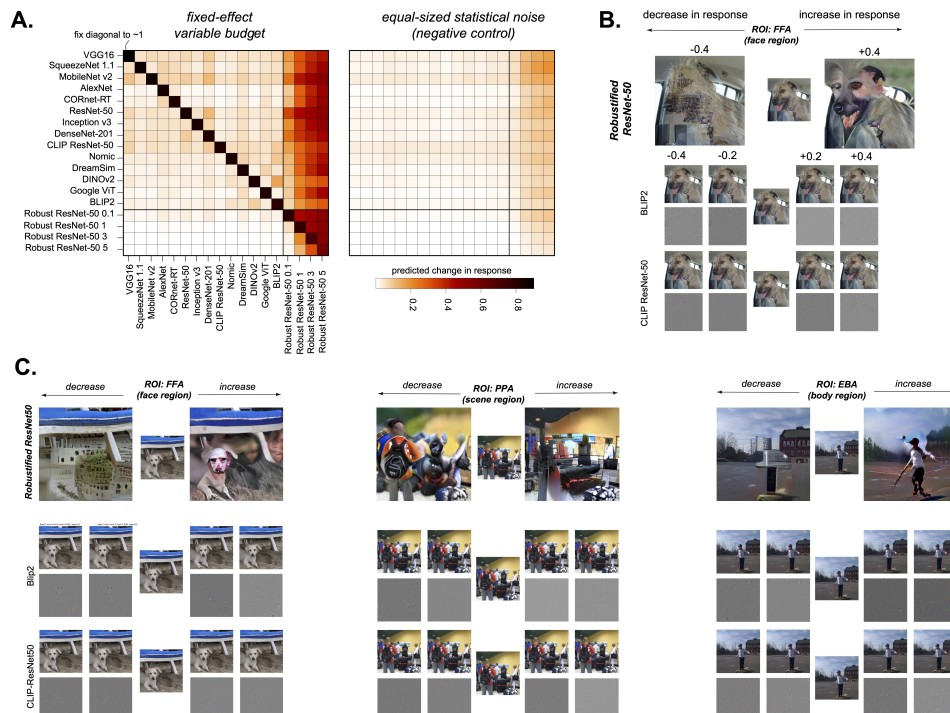

Figure 4: **Robustified models generate generalizable and interpretable adversarial probes**
**A:** Transfer matrix illustrating the sensitivity of a model (y-axis) to adversarial perturbations generated on a source model (x-axis). Here, the diagonal is fixed to be approximately 1.0, and the perturbation budget $\epsilon$ is increased until reaching this threshold. Displayed to the right of this transfer matrix is the equal-sized statistical noise, representing the negative control. **B:** Example plots visualizing the perceptual effect to the image. On the top row, perturbed images are visualized from robustified ResNet-50. To the left, images are predicted to substantially decrease the response in the FFA, downplaying face-like features. Images to the right are predicted to maximize FFA response, emphasizing face-like features. The second row and third rows respectively depict the same for BLIP2 and CLIP-ResNet50. For these models, a significantly smaller $\epsilon$ is necessary to reach the desired $s_i$, so these images are not perceptually informative. **C:** Three more examples, comparable to (B). Here, visualizations are shown for modulating brain responses in the FFA, PPA, and EBA respectively.

measured the subspace membership by asking how much of model $i$'s leading direction lies within model $j$'s subspace and measuring the projection energy $\|\mathcal{P}_j \mathcal{P}_j^\top v_{i,1}\|_2^2$ (where $v_i$ are the singular vectors for model $i$). Finally, we measured the full subspace overlap between model $i$ and $j$ as the average cosine of the principal angles $\{\theta_l\}_{l=1}^k$ between them, $\frac{1}{k}\sum_{l=1}^k \cos(\theta_l)$. These three metrics range from 0 (orthogonal subspaces) to 1 (identical subspaces). Further details on perturbation subspaces, along with their relationship to adversarial attacks, can be found in A.4.

Figure 3E summarizes subspace similarity results across models for all three metrics. Across all analyses, equal predictivity did not imply shared representational geometry. Distinct brain models occupied largely distinct perturbation subspaces. The cosine similarity of their leading axes was near zero, the mean cosine of principal angles between their subspaces was low, and the projection energy of one model's top direction into another's subspace was minimal. The input directions that most potently modulated one model's responses were largely independent of those that affected other models. Consistent with the single-direction analysis, robustified models showed a different pattern. Their perturbation subspaces exhibited greater overlap with each other but showed only modest alignment with those of standard models. This reinforces the conclusion that while standard ANN models achieve high brain prediction scores via distinct and idiosyncratic coding axes, robust training (robustified ANN models) partially regularizes and aligns these sensitive subspaces.

## Section 4: Robustified models enable transferable and semantically meaningful adversarial probes

An ideal brain model should function as an *in silico* experimental testbed allowing us to generate new targeted hypotheses (images) about neural representations. In this section, we pursue this goal directly by asking which specific models would enable us to design minimal, interpretable perturbations that change the neural response, or "*small-norm neural guidance*". Our earlier analyses used small, fixed-budget perturbations to characterize each ANN-based model's local representation. But as we have seen in Figures 1 and 3, this approach has limits for neural guidance studies. For standard models, the adversarial probes were uninterpretable noise (Figure 3C top-left); for robust models, they were too weak to even induce a significant change in model's response (Figure 3C top-right). We therefore inverted our logic. Instead of fixing the perturbation budget, we fixed the target sensitivity for each model (the diagonal in Figure 3B) and allowed the perturbation size to vary. This shift served two goals. First, it allowed us to test the key hypothesis that the most efficient path to altering a brain model's prediction is through a semantically meaningful change to the image, not random noise. If true, small-norm perturbation probes would be interpretable and the brain models would directly reveal the specific visual features to which a given brain region is most sensitive. Second, it allowed us to explore whether adversarial probes can be used as tools for neural guidance. The optimized images are candidate stimuli hypothesized to change brain responses intended for use in subsequent human experiments. By synthesizing optimized images that drive predicted changes in neural responses, these methods provide candidate stimuli for future experiments aimed at identifying causal "knobs" of visual representation in the brain.

Our "fixed-effect, variable-budget" analysis revealed two key findings. First, consistent with their design, robustified models required substantially larger perturbations to achieve the target effect size. Despite this higher "cost," the adversarial probes generated on these models were highly effective, reliably transferring to other models, especially other robust architectures (Figure 4A). We quantified the magnitude of this transfer using a Control-Consensus Score (CCS), defined as the mean predicted change in response in all other models normalized by the predicted response change in the source model:

$$\text{CCS}(i) = \frac{1}{M-1} \sum_{\substack{j=1 \\ j \neq i}}^{M} \frac{\Delta r_j}{\Delta r_i}$$

This metric directly quantifies how consistently a perturbation designed to change the response in one model is predicted to change responses in other models. A CCS close to 1 means the models share a common coding axis; a CCS near 0 means the perturbation direction is model-specific and does not generalize. We find that robustified models have consistently higher CCS scores than all baseline models. This turns local geometry alignment into a brain-relevant metric of consensus feature coding. We find that sensitivity and CCS show an inverse association (Pearson $R = .77$, all models). As a model becomes more sensitive, its perturbations are less likely to transfer to other models. This association holds to a lower degree ($R = -.40$) even when we remove the adversarially trained models. Note however that the relation is better captured by a logistic regression ($R^2 = 0.98$) than a linear association. Model values for both sensitivity and CCS are reported in **??**

This effect cannot entirely be explained by the perturbation magnitude alone (see control with equal-sized statistical noise). What do these stimuli look like and can we use them to discriminate between candidate models of the brain? Probes generated from robustified models consistently produced semantically interpretable changes to the input image, aligned with the known function of the target brain region. For instance, adversarial probes targeting the fusiform face area (FFA) systematically transformed an image to appear more or less "face-like" (Figure 4B). Similarly, probes designed to increase the parahippocampal place area (PPA) response altered images by converting people into background elements, while probes designed to decrease the response would blur or erase scene components entirely (Figure 4C). Following the same logic, probes for the extrastriate body area (EBA) selectively emphasized or removed body parts.

Even the most robust standard model (BLIP2) did not change the image, a very systematic effect. These images represent strong, testable predictions about the causal features that drive these brain regions, which we aim to verify in future neuroscience experiments. At a minimum, the current

results establish that our method is a powerful generative tool, capable of producing the targeted, hypothesis-driven stimuli necessary for such causal tests.

## 4 DISCUSSION AND LIMITATIONS

In this study, we systematically characterized the local representational geometry of ANN-based brain models using targeted adversarial probes. We first showed that standard models, though highly predictive of neural responses, are unexpectedly fragile: small, imperceptible perturbations reliably disrupted their responses, marking a clear divergence from the brain (Section 1). We then demonstrated that adversarial sensitivity provides a sharper criterion than predictivity for distinguishing between candidate brain models (Section 2), and that standard models occupy distinct, non-transferable perturbation subspaces (Section 3). By contrast, robustified models were more stable, their perturbations transferred more readily across models, and the changes they produced aligned with the known selectivities of high-level visual regions (Section 4). We introduced a **control-consensus score**, quantifying the tendency for adversarially trained model's perturbation probes to effectively modulate other models. Our contributions are threefold. Conceptually, we introduce the idea of the local coding axis as a principled criterion for discriminating between brain models. Methodologically, we adapt adversarial probes, traditionally used to expose model weaknesses, into a neuroscience tool for characterizing and comparing local representational geometries. Scientifically, this framework provides evidence that robustified models are better candidates than standard networks for capturing brain-like representations. Finally, by turning these probes into a generative tool, we pave the way for targeted stimuli that can directly test causal hypotheses in future vision neuroscience experiments. This work aligns with a growing literature suggesting that adversarially trained models are more "human-like", as shown using visual and auditory metamers (Feather et al., 2023), controversial stimuli (Skrill et al., 2025), and configural shape scores (Doshi et al., 2025). We note that both prediction and sensitivity measure different things (global and local alignment respectively) and believe it is necessary to evaluate both complementary metrics when considering brain-model alignment.

There are several ways in which adversarial probes could lead to new neuroscientific discoveries in future work. First, adversarial probes in robustified models could be used to identify the attributes that drive neural responses most strongly in a given brain region. Low-norm stimuli are particularly useful, because they keep the stimulus features close to the original stimulus and can effectively identify very specific features without the need for more complicated structural or naturalistic priors. A second direction is reverse-engineering stimuli without naturalistic generative priors. Current tools in human neuroscience rely on generative models (Luo et al., 2023; Cerdas et al., 2025; Ratan Murty et al., 2021) which inject strong priors about the generative model's training distribution. In contrast, adversarial probes operate directly on the local coding axis of the encoding model and do not require a generative prior. This opens the door for non-trivial discoveries that are currently inaccessible, such as identifying feature directions to which the brain is sensitive but do not correspond to any human-interpretable dimension.

In addition, our work is closely related to testable (and falsifiable) predictions about neural activity. First, we expect that very small-scale perturbations which affect models should have no measurable effect on human fMRI responses. Second, adversarial probes generated from robustified models (and not baseline models) should reliably modulate neural responses in the human brain. The control-consensus score also provides a falsifiable prediction regarding the degree of transfer to the brain for robustified models.

**Limitations:** Our study has three main limitations. First, our conclusions are based on small-scale representational geometry. While we show that robustified models better capture local coding directions, it remains an open question whether other types of models might more accurately capture large-scale representational structures in the brain. A full account of brain-like computation will likely require integrating both local robustness and global organization. Second, our claims are analytical and computational. Although we generate concrete predictions about neural coding, these must be validated in new neuroscience experiments. Finally, while we argue that the representations of robustified models are more brain-like, we make no claims about how robustness arises in the brain. The biological mechanisms that produce robustness may differ from adversarial training, and clarifying these processes remains an important goal for future work.

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

# A  APPENDIX

## A.1  DETAILS ON ENCODING MODELS

**Model architectures**: We considered 14 pre-trained artificial neural network architectures previously validated against brain data. These included eight convolutional neural networks (ResNet-50 (He et al., 2016), VGG16 (Simonyan and Zisserman, 2015), Inception v3 (Szegedy et al., 2016), SqueezeNet v1.1 (Iandola et al., 2016), AlexNet (Krizhevsky et al., 2012), CORnet-RT (Kubilius et al., 2018), DenseNet201 (Huang et al., 2017), MobileNet v2 (Sandler et al., 2018)), three self-supervised vision transformers (DINOv2 (Oquab et al., 2023), DreamSim-ViT-B/32 (Fu et al., 2023), Google ViT (Dosovitskiy et al., 2021)), and three vision–language models (CLIP-ResNet50 ((Radford et al., 2021)), BLIP2 ((Li et al., 2023)), Nomic (Nussbaum et al., 2024)). We additionally used publicly available adversarially trained models (Engstrom et al., 2019; Ilyas et al., 2019). For $l_2$-bounded attacks, we evaluated models trained with $\epsilon = 0.1, 1, 3, 5$, and for $l_\infty$-bounded attacks, we evaluated models trained with $\epsilon = 0.5/255, 1/255, 2/255, 4/255, 8/255$.

**Encoding model cross-validation procedure:** We used 1000 shared images across four subjects from the NSD dataset, of which 515 were also viewed by an additional four subjects. In our study, these 515 images served as a held-out test set for all evaluations, while the remaining 485 images were used for training and validation.

Each neural network architecture comprises multiple layers whose activations provide candidate representations for encoding models. To determine the optimal set of representations, we constructed linear encoding models for each subject and brain region, selecting the layer that yielded the highest average cross-validated predictive accuracy across voxels. We focused on the second half of layer representations, as previous work has shown the optimal layer for predicting high-level visual cortex voxels to be downstream in the architecture. For each layer, we applied ridge regression, with the regularization parameter strength chosen via cross-validation from ten logarithmically spaced values between 1e-2 and 1e6, optimizing for maximum predictive accuracy (by correlation).

**Encoding Model Discriminability** We evaluate the ability of both metrics (adversarial robustness and model predictivity) to discriminate encoding models of the brain. For each of the eight models evaluated, we compute the average sensitivity across all subjects and brain regions. We explore whether the spread of the adversarial robustness distribution of the encoding models will be greater than the spread of the model predictivity distribution (i.e., "adversarial robustness" serves as a better discriminative tool). To evaluate this, we test the variance and sparseness of both adversarial sensitivity and predictivity.

- **Normalized Variance:** Since the scale of "sensitivity" (unbounded) and "predictivity" (bounded $-1$ to 1) are different, we cannot directly compare the variances. Instead, we first divide all accuracy and sensitivity values by their respective maximum value before reporting the variances (hence normalized variance).

- **Sparseness:** We use the sparseness metric defined in (Willmore et al., 2011; Vinje and Gallant, 2000). Specifically, for a distribution of values $P(r)$, sparseness (S) is computed with the following:

$$S = 1 - \frac{E[r]^2}{E[r^2]},$$

where $E[\cdot]$ denotes the expectation operator.

## A.2  DETAILS ON THE ADVERSARIAL ATTACKS

We consider two variants of adversarial attacks that perturb an input image $x$ to change a single model output $r$ while keeping the perturbation small.

$\ell_\infty$-**based attack.** We keep a per–channel-bounded perturbation $\delta$ with $\delta_c \in [-\epsilon_c, \, \epsilon_c]$. At each step we take a signed gradient step on the objective

$$\mathcal{L}(x + \delta) = \begin{cases} -f(x+\delta)_i & \text{to minimize } f(x)_i, \\ f(x+\delta)_i & \text{to maximize } f(x)_i, \end{cases}$$

and clip back to the $\ell_\infty$ box:

$$\delta \leftarrow \text{clip}_{[-\epsilon, \, \epsilon]}\big(\delta + \alpha \, \text{sign}(\nabla_\delta \mathcal{L})\big).$$

After $T$ steps we form the adversarial image $x^{\text{adv}} = \text{clip}_{[\min,\max]}(x + \delta)$. When $T{=}1$ and $\alpha{=}\epsilon$, this reduces to FGSM (Goodfellow et al., 2015).

$\ell_2$-**based attacks.** Here, $\epsilon$ and $\alpha$ are scalars and the perturbation is constrained by $\|\delta\|_2 \leq \epsilon$. Each step takes a normalized gradient step and (if needed) projects back to the $\ell_2$ ball:

$$\delta \leftarrow \delta + \alpha \, \frac{\nabla_\delta \mathcal{L}}{\|\nabla_\delta \mathcal{L}\|_2 + \eta}, \qquad \delta \leftarrow \begin{cases} \delta & \text{if } \|\delta\|_2 \leq \epsilon, \\ \delta \cdot \dfrac{\epsilon}{\|\delta\|_2 + \eta} & \text{otherwise,} \end{cases}$$

with the same final clipping $x^{\mathrm{adv}} = \mathrm{clip}_{[\min,\max]}(x + \delta)$. A small $\eta > 0$ provides numerical stability when the gradient norm is near zero.

We set $\epsilon = 5$ and $\epsilon = 3/255$ for the $l_2$- and $l_\infty$-bounded attacks respectively. We note that these are related (and empirically, we observe they are approximately equal) due to the the norm inequality

$$\|\delta\|_2 \leq \sqrt{p}\|\delta\|_\infty.$$

Since our images have $p = 224 * 224 * 3$ pixels, an $l_\infty$ budget of $\epsilon = 3/255$ corresponds to a worst-case $l_2$ norm of $\sqrt{224 * 224 * 3}(0.012) \approx 5$.

### A.3   RESULTS ON $L_\infty$-BOUNDED ATTACKS

In this study, we conducted both $l_2$- and $l_\infty$-bounded attacks for all analyses. Unlike an $l_2$-bounded attack, which can appear to concentrate the noise pattern on the salient parts of an image, an $l_\infty$-bounded attack constrains every pixel to change by at most $\epsilon$. This means the perturbation is spread out uniformly: instead of a few pixels changing a lot, all pixels are adjusted by small amounts.

We find that results on $l_\infty$-bounded attacks are highly consistent with $l_2$-bounded attacks, suggesting that our results are not dependent on the exact parameters and implementation of the adversarial attack. Notably, the voxelwise results (over all models, subjects, and regions) from the $l_\infty$-bounded attacks are highly correlated with the results from the $l_2$-bounded attacks ($R$=.97, $P$ <0.00001). We do note, however, that the differences between the two attacks are subtly reflected in the ranking of models by sensitivity (Figure 5A): the exact order of models is slightly different between the $l_2$ and $l_\infty$ attacks. The general trend of models is consistent within both ranks.

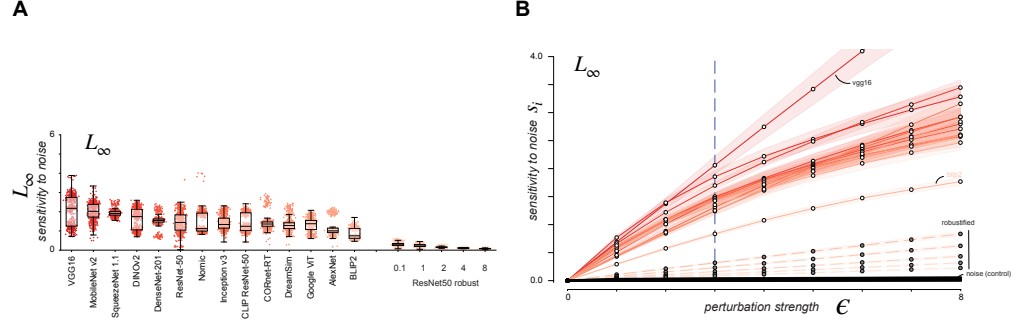

Figure 5: **Adversarial sensitivity provides a sharper test of brain models than predictivity.** **A:** Boxplots showing the predictive accuracy of candidate encoding models (x-axis) against brain responses (y-axis). Boxes indicate the median prediction accuracy with error bars for voxel-level standard error; dots correspond to individual voxels. Bottom: Boxplots showing adversarial sensitivity (y-axis) for the same models, measured at a perturbation budget of $\epsilon = 5$. **B:** Adversarial sensitivity functions. The x-axis indicates perturbation strength ($\epsilon$) and the y-axis shows model sensitivity. Lines represent different models. Standard models (solid) exhibit steep increases in sensitivity even at very small perturbations (e.g., VGG16), whereas robustified models (dashed) are more stable, requiring larger perturbations to shift predictions. The black control line shows randomized noise, which has minimal effect.

Like in the case of $l_2$-bounded attacks, we observe that non-adversarially trained models exhibit steep increases in sensitivity even at very small increases of $\epsilon$, whereas robustified models remain more stable (Figure 5B).

In addition, we replicate the results in Sections 3 and 4 (Figure 6). We find that perturbations for a model generated under the $l_\infty$ constraint generally do not transfer to other models. When fixing the target sensitivity for each model instead of the $\epsilon$ budget, we again find that 1) robustified models require larger perturbations to

achieve the target effect size, and 2) adversarial probes generated on the robust models reliably transfer to other models (including the other robust models).

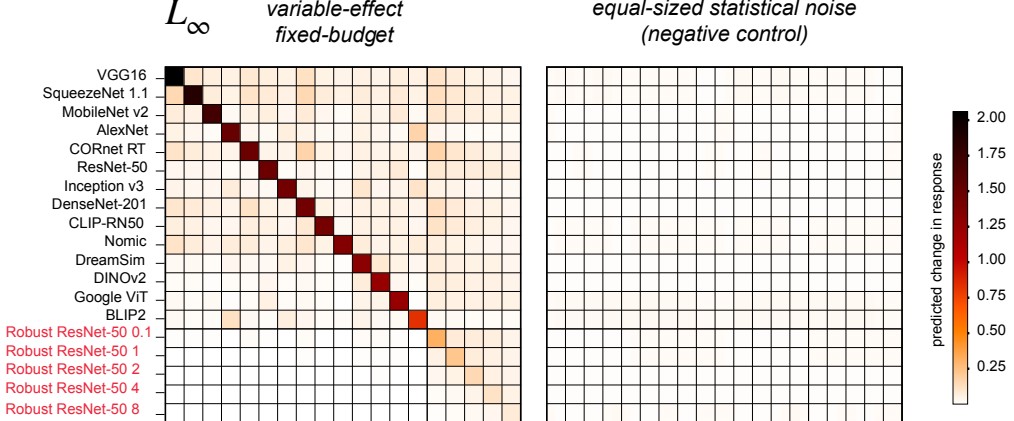

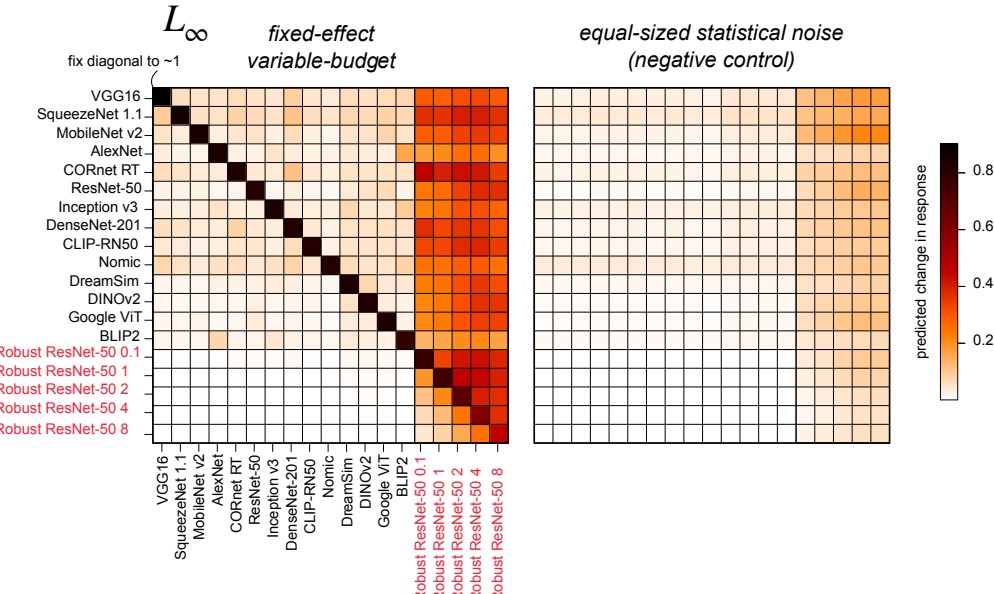

Figure 6: Top: transfer matrix of adversarial probes ($\epsilon = 3/255$). Strong self-effects (diagonal) contrast with weak transfer across models, especially among standard networks. Bottom: Transfer matrix illustrating the sensitivity of a model (y-axis) to adversarial perturbations generated on a source model (x-axis). Here, the diagonal is fixed to be approximately $1.0$, and the perturbation budget $\epsilon$ is increased until reaching this threshold. Displayed to the right of this transfer matrix is the equal-sized statistical noise, representing the negative control.

## A.4 DETAILS ON PERTURBATION SUBSPACES

We formalize the notion of a perturbation subspace as follows. Consider an image $x \in \mathbb{R}^{C \times H \times W}$. Flattening the image gives a vectorized representation $x \in \mathbb{R}^p$, where $p$ denotes the total number of pixels. This image is passed through our representational encoder $f$ and linear readout $g$ to produce a predicted response vector $r = g(f(x)) \in \mathbb{R}^m$, where $m$ is the dimensionality of the neural data being predicted (in our study, the number of voxels in a given subject and region).

To analyze how infinitesimal changes in the image affect $r$, we study the Jacobian of voxel predictions with respect to input pixels,

$$J = \partial r/\partial x \in \mathbb{R}^{m \times p}.$$

For a sufficiently small perturbation $\delta$, the predicted response satisfies the Taylor expansion $r(x+\delta) = r(x) + J\delta + \frac{1}{2}\,\delta^\top H(x+\theta\delta)\,\delta$, where $H$ is the Hessian matrix. To first order, we have $\Delta r = J\delta$.

The effect of a perturbation $\delta$ on the responses is determined by the quadratic form,

$$\|\Delta r\|_2^2 \;=\; \delta^\top (J^\top J)\,\delta,$$

associated with the symmetric positive semidefinite matrix $J^\top J \in \mathbb{R}^{p \times p}$, which encodes how strongly different directions in pixel-space influence the magnitude of the voxel-response change. The eigenvalues measure the strength of this influence, and the eigenvectors identify the corresponding directions in pixel space. The top-$k$ eigenvectors of $J^\top J$ (equivalently, the top right singular vectors of $J$) span the perturbation subspace $\mathcal{P} \in \mathbb{R}^{p \times k}$. For the analyses in this study, we set $k = m$ (the number of voxels).

**Relation to adversarial attacks.** Perturbation subspaces characterize the directions in pixel space that most strongly modulate the multi-voxel response vector. For a perturbation $\delta$ with $\|\delta\|_2 \leq \varepsilon$, $\sigma_1$ (the leading singular value of $J$) is the optimal attack to maximize the total energy in the voxel-response change.

In contrast, voxel-wise adversarial attacks maximize the change for a single output coordinate. Locally, $r_i(x+\delta) \approx r_i(x) + g_i^\top \delta$, where $g_i$ is the gradient for voxel $i$, $g_i = \nabla_x r_i(x)$. In this case, the first-order optimal perturbation is

$$\delta_i^\star = \varepsilon \frac{g_i}{\|g_i\|_2}, \qquad |r_i(x + \delta_i^\star) - r_i(x)| \;\approx\; \varepsilon \|g_i\|_2.$$

Comparing the two, the multi-voxel vector optimal direction achieves

$$|g_i^\top v_1| = \|g_i\|_2\,|\cos\phi_i|,$$

where $\phi_i$ is the angle between $g_i$ and $v_1$. As a result, the voxel-wise optimum is always at least as strong for that voxel (achieving the full $\varepsilon\|g_i\|_2$), while the subspace optimum may be strictly weaker by a factor $|\cos\phi_i| \leq 1$. A similar derivation follows for $L_\infty$-bounded attacks. It is important to note, however, that this comparison is a first-order analysis, assuming linearity of the model (valid for infinitesimal perturbations). For finite and larger $\epsilon$, however, it is possible that higher-order terms will significantly alter both the optimal direction and the achieved change due to the nonlinearity of the model. As a result, the relationship between voxelwise sensitivities $s_i$ and subspace directions is approximate and may break down in strongly nonlinear regions. We use perturbation subspaces mainly as a geometric probe of local representational sensitivity, not as a literal predictor of global attack strength.

## A.5 STATISTICS FOR SENSITIVITY AND CONTROL-CONSENSUS SCORE

| Model | Sensitivity | Control Consensus Score |
|---|---|---|
| VGG16 | 2.9472 | 0.0341 |
| SqueezeNet v1.1 | 2.5658 | 0.0219 |
| MobileNet v2 | 2.4029 | 0.0268 |
| AlexNet | 1.9280 | 0.0376 |
| Inception v3 | 1.7963 | 0.0240 |
| DenseNet 201 | 1.7947 | 0.0509 |
| CORnet_RT | 1.7827 | 0.0317 |
| ResNet-50 | 1.7799 | 0.0360 |
| CLIP-RN50 | 1.7502 | 0.0177 |
| Dino v2 | 1.7382 | 0.0276 |
| Nomic | 1.6463 | 0.0274 |
| Dreamsim | 1.6396 | 0.0429 |
| Google ViT | 1.5630 | 0.0369 |
| Blip 2 | 0.7632 | 0.0472 |
| L2-RN50-robust-0.1 | 0.9876 | 0.1547 |
| L2-RN50-robust-1 | 0.4283 | 0.3530 |
| L2-RN50-robust-3 | 0.2793 | 0.5311 |
| L2-RN50-robust-5 | 0.1988 | 0.7372 |

