# OpenReview forum: "Targeted perturbations reveal brain-like local coding axes in robustified, but not standard, ANN-based brain models"
_ICLR.cc/2026/Conference — Submitted to ICLR 2026_

### Official Review · Reviewer_Qijk · 2025-10-31

**Soundness:** 4
**Presentation:** 3
**Contribution:** 2
**Rating:** 4
**Confidence:** 5

**Summary:**

This study uses adversarial perturbation methods to investigate how different brain models local representational geometry. The paper presents four main findings. First, even when different models achieve similar voxel-response prediction accuracy, their local geometries differ substantially and are highly sensitive to adversarial perturbations. This suggests that adversarial sensitivity may be a more informative indicator than prediction accuracy. Second, different models represent local geometry in distinct subspaces. Finally, the authors show that robustified models can better explain neural representations in the brain.

**Strengths:**

1. This study creatively applies adversarial perturbation to examine the local geometry of neural representations, which, to my knowledge, has not been systematically explored before.
2. The overall methodology and analyses are solid with no obvious technical flaws.
3. The writing is clear, and the logical flow is easy to follow.

**Weaknesses:**

1.	The figures are poorly prepared—the text is barely legible, and the resolution is low. This seriously affects readability.
2.	The core logic of the argument could be reconsidered to strengthen the conceptual contribution.
3.	In Section 4, the rationale for how new stimuli can be used to test neuroscience theories is not clearly articulated.

**Questions:**

1. The study is very interesting and provides a valuable example of how brain representations of images can be analyzed through adversarial perturbations. However, the results are not entirely surprising. The voxel encoding model linearly maps ANN features to voxel responses. Therefore, the adversarial sensitivity of predicted brain responses should largely depend on the adversarial sensitivity of the ANN features themselves. Since we already know that standard ANNs (without adversarial training) are fragile, it follows naturally that the voxel responses derived from them would also be fragile.

2. In Section 2, the authors claim that adversarial sensitivity distinguishes between models better than prediction accuracy. Indeed, Figure 2A seems to support this. However, it is unclear why higher sparseness indicates that adversarial sensitivity is a better metric—the theoretical justification for this link should be elaborated.

3. From Figure 2A, BLIP2 appears to perform better in terms of adversarial sensitivity. Why is that the case? Can we therefore conclude that BLIP2 is a better brain model overall? If so, what is the underlying reason for this advantage?

4. In Section 4, the authors suggest using adversarial images as new stimuli to probe which features different brain regions respond to. However, the examples provided remain quite intuitive (e.g., that FFA responds more to face-like features) and do not yet seem to offer novel neuroscientific insights. Further discussion on how this approach could generate non-trivial discoveries would strengthen the paper.

---

> ### Author Response · Authors · 2025-11-25
> **Response 1 to Reviewer Qijk**
>
> We thank the reviewer for their close reading and constructive engagement with our work. We are especially grateful for the reviewer’s note that local representational geometry has not been systematically explored before. We fully agree. We want to also emphasize that our study also demonstrates *why* this level of analysis matters especially in relation to several of the reviewer’s thoughtful concerns. Below, we take each remaining point head-on. Doing so has substantially strengthened our paper.
>
> **W1. Figure Quality**
>
> Thank you for bringing this up, but we are genuinely surprised by this reviewer comment. This is because all our figures were exported as vectorized, high-resolution PDFs. To double check, we re-downloaded our submission from OpenReview, and the file is 37MB almost entirely because the figures are embedded at very high resolution. None of our figures pixellate when zoomed in.
> Moreover, two other reviewers have explicitly praised the clarity of our figures.
>
> (Reviewerr auvF): “good figures for the readers to follow”
> (Reviewer SpWp): “the figures are very well done to clearly illustrate a few ambiguous concepts”
>
> This makes us think that there may have been a display/rendering issue on Reviewer’s end (eg. browser PDF renderer?). We are sorry that the Reviewer faced readability issues, but we want to assure the Reviewer that all our figures (in the original, the re-submission, and final camera-ready version) are fully vectorized, publication-quality.
>
> **W2/Q1: Clarifying the core logic of the arguments**
>
> We appreciate the reviewer’s comment, and we also understand why the first part of our results may appear unsurprising at a glance. However, we want to underscore that this is only the first of the three distinct facets of our study each building up to a much larger (non-intuitive and to our knowledge previously unstated) picture.
>
> 1. We use part 1 to establish an important diagnostic point: prediction alone is insufficient as a metric of model goodness. Even state-of-the-art predictive models can be extremely fragile in neural prediction space. This section is important for the big-picture store because it reveals a core limitation of current evaluation paradigms and shows that model predictions for even several state-of-the-art brain models are predictive.
> 2. In part 2, we go beyond and show what the fragility actually *means* in the representational sense. Fragility exposes differences in the local-coding axis (the first-order representational geometry) of each model. Our contribution is to clearly show that models with nearly identical prediction scores can nevertheless learn different local representational spaces. This phenomenon has not been documented before, and it matters because local coding structure is what governs how models extrapolate and is important for neural control (as we show next).
> 3. Part 3 demonstrates that this difference in local geometry has real-world consequences. Previously, this section remained qualitative. Based on this feedback and others (Reviewer ukS1 and others), we now include a Control Consensus Score (CCS) metric, which quantifies how consistently the perturbation probe for a given model predicts change in other models. We find that robustified models have higher CCS (in large part because of their smooth local representational spaces). Their local coding structure better aligns with other models and is also more semantically meaningful.
>
> To summarize, our study unfolds through three conceptual stages that build a coherent framework for understanding the local representational geometry of brain models. We (1) expose the limits of prediction, (2) formalize the idea of local-coding geometry and (3) show concretely that differences in local-coding geometry are associated with model prediction robustness and matter for closed-loop control studies. Part 1 (which the Reviewer references in their response) is what exposes a key issue and motivates the need for the conceptual and methodological contributions in parts 2 and 3. We respectfully ask the reviewer to view the work in its full arc rather than only the initial observation of fragility.

---

> > ### Author Response · Authors · 2025-11-25
> > **Response 2 to Reviewer Qijk**
> >
> > **Q2. Why does high sparseness between models matter?**
> >
> > Thanks for raising this question, and we fully agree that the theoretical justification should be elaborated further.
> >
> > In this section, we were responding to a commonplace complaint in NeuroAI which is that several models seem to be saturating. The implicit (often unstated) goal of these benchmarks is to obtain *some* discriminability between models so that we can evaluate which models better capture brain-like computation.
> >
> > Now we start by demonstrating that models predictions might be high across models but that those predictions tend to be fragile for some (and not all models). Sparseness (which is a commonplace metric in non-human primate neuroscience) is an effective metric to test the discriminability between models. We admit that it is non-standard in the model-comparison domain hence we added a more intuitive normalized variance measure alongside it.
> >
> > Discriminability between models seems to be important for those who want to benchmark models on brain-likeness (though we acknowledge that not everyone in the field prioritizes this equally).
> >
> > **Change:** We will add a short subsection expanding on the theoretical justification for why sparseness between models matters for model benchmarking
> >
> > **Q3. Is BLIP2 (or other models better) a better model of the brain based on Figure 2A?**
> >
> > The Reviewer raises an important question which is very much worth clarifying.
> >
> > Based on Figure 2 we can claim that model predictions from BLIP2 are less susceptible to small-scale adversarial noise. How sensitive is the brain? Our estimate is that the brain at this scale will be at 0 (note the perturbation strengths were deliberately kept lower than the resolution of the human eye in this stage). By this measure it is also fair to say that BLIP2 is closer to the brain’s estimated sensitivity (which is 0).
> >
> > As we elaborate above (response to W2/Q1), we take the analysis further by asking a more stringent and neuroscientifically relevant question: Do adversarial probes generated from BLIP2 generalize to other models? Our newly included Control Consensus Score (CCS) provides exactly this comparison. Here we find that BLIP2 in fact does not generalize well. Its CCS is only 0.05. This means that if the adversarial probe predicted to drive a unit change in BLIP2’s own response, it changes the response in other models by only 5% of that amount. So BLIP2 might appear better on one dimension (local stability) but weaker on another (shared local-axis geometry).
> >
> > Stepping back, we think it is irresponsible to equate *any* single metric as an absolute test of brain-likeness. As we emphasize throughout the rebuttal, these comparisons are independent and complementary. A strong model of the brain should be required to pass all of these tests. None of them alone is decisive. The goal of our study is to shine light on model robustness, and the shared local-axis similarity across models.

---

> ### Author Response · Authors · 2025-11-25
> **Response 3 to Reviewer Qijk**
>
> **Q4/W2. How can adversarial probes lead to non-trivial neuroscientific discoveries?**
>
> We agree the need to further discuss on how to use adversarial probes to generate non-trivial discoveries. We also fully agree that the examples shown in Section 4 (e.g., FFA moving toward face-like structure) are intuitive. But this also exactly why we chose these brain areas. They serve as good sanity checks that our framework works on well-known cases. The goal of this section is not to claim new neuroscientific results but to demonstrate the feasibility of using local adversarial probes to systematically manipulate predicted brain responses and to showcase that only some models generate semantically meaningful changes to the image.
>
> We think there are least 2 ways to take this further (that we are personally excited about):
>
> 1. **Revealing specific features that drive responses.** For example, “what it is exactly about the face that drives responses in the FFA”.  Low-norm stimuli are great because they keep the stimulus features close to the original stimulus and can identify very specific features without the need for more complicated structural or naturalistic priors (like BrainActiv for example).
> 2. **Reverse-engineering stimuli without naturalistic priors.** The current tools in human neuroscience use generative models (GANs or Diffusion models) which inject strong priors about the generative model’s training distribution. In contrast, adversarial probes operate directly on the *local coding axis* of the encoding model and do *not* require a generator, latent codes, or naturalistic structure. This opens the door for non-trivial discoveries that are currently inaccessible, such as identifying feature directions that the brain is sensitive to but that do not correspond to any human-interpretable dimension or any generator-prior axis.
>
> **Changes:** We will do as the Reviewer themselves suggested and will expand this discussion and clearly articulate aspects of our response. We will also concrete examples of how adversarial perturbations could be used to learn something new about the brain in future work.
>
> We sincerely thank the reviewer for their constructive engagement with our work. We have directly and comprehensively addressed each concern. We expanded the theoretical justification of our metrics, added new analyses (such as the Control Consensus Score), clarified neuroscientific predictions, and strengthened the discussion of future neuroscientific applications. These changes have further improved the rigor and clarity of our work.
>
> We think the paper is comprehensive and rigorous and introduces a principled way to study local representational geometry (as aspect that, as the reviewer themselves noted, has not been systematically investigated).  We respectfully ask the reviewer to reconsider their overall assessment of the paper in light of these clarifications. Please do not hesitate to reach out if there is anything else we can clarify or improve.

---

> > ### Comment · Reviewer_Qijk · 2025-11-27
> >
> > I appreciate the authors' response.
> >
> > I did not see that the authors directly my first question
> > The study is very interesting and provides a valuable example of how brain representations of images can be analyzed through adversarial perturbations. However, the results are not entirely surprising. The voxel encoding model linearly maps ANN features to voxel responses. Therefore, the adversarial sensitivity of predicted brain responses should largely depend on the adversarial sensitivity of the ANN features themselves. Since we already know that standard ANNs (without adversarial training) are fragile, it follows naturally that the voxel responses derived from them would also be fragile.
> >
> > I have no problem with the results here. My feeling is that this metric is not novel as it is very intuitive because most the models per se is fragile. so prediction geometry is naturally fragile.

---

> > > ### Author Response · Authors · 2025-11-27
> > > **Followup 1 to Reviewer Qijk**
> > >
> > > We thank the Reviewer for engaging in this discussion. We are very glad to no there are no remaining concerns about the correctness of the results themselves.
> > >
> > > We’d like to respectfully note that we did attempt to directly address the question under section W2/Q1 where we acknowledged that Part 1 may appear intuitive in isolation. However, because this point is central, we agree that it deserves its own discussion and would like to tackle it directly here.
> > >
> > > We want to start with the point of agreement. *We agree* with the reviewer’s general intuition. Because voxel encoding models are linear readouts of ANN features, one would expect neural predictions to inherit some degree of the ANN’s adversarial fragility. This part of the result might be intuitive (subjectively, the intuition itself is variable among peers).  Our goal here was not to claim novelty in the fact that predictive models can be fragile (and we have been careful not to), but to make a clear and necessary point: prediction alone is not a sufficient test of brain-likeness. There is, as the reviewer notes, a relatively “easy” way to break even highly predictive models. Exposing that failure mode is important precisely because this fragility is invisible to standard benchmarks.
> > >
> > > However **this observation is the first and most superficial layer of the study**.  This section is an unavoidable pitstop in a much longer conceptual arc. This part simply exposes the issue and motivates the deeper scientific question (How do we characterize the local-coding geometry of brain models?). We implore the Reviewer to consider the sections beyond.
> > >
> > > **Our key findings come after this initial observation.** We show that models with low s_i exhibit several non-trivial properties:
> > >
> > > 1. They converge on shared local perturbation geometry across models.
> > > 2. They exhibit semantically structured principal directions in their sensitivity subspaces.
> > > 3. They achieve high Control Consensus Scores (CCS), meaning their perturbations generalize better across other models.
> > >
> > > Note that we find that low sensitivity *alone* does not explain these properties. For example, BLIP-2 is highly robust under the s_i metric, yet it does **not** show strong transferability or semantic alignment (as the Reviewer very astutely pointed out) and has a very low CCS. Hence these properties (structured robustness) are not guaranteed by robustness and are not trivial consequences of linear readout.
> > >
> > > To be clear, this is precisely why out title and Discussion do not frame the paper as being about adversarial sensitivity. The paper is about local-coding axes in brain models. The fragility result is an entry point to this finding.
> > >
> > > The scientific contribution lies in demonstrating that (1) models with similar predictivity differ systematically in *local geometry* (2) that this geometry has structure (limited linear association with adversarial sensitivity), and (3) that the local-coding structure matters for closed-loop generalization (neural control).
> > >
> > > We respectfully ask the reviewer to evaluate the paper based on its full arc (esp new CCS metric that helps expose this idea better) rather than on the initial diagnostic observation alone. Given the rigor of the work and the clarifications and analyses added in response to this review, we also respectfully request a renewed consideration of the overall assessment.

---

### Official Review · Reviewer_auvF · 2025-10-31

**Soundness:** 3
**Presentation:** 3
**Contribution:** 1
**Rating:** 2
**Confidence:** 4

**Summary:**

The paper propose to use adversarial examples as a method to probe the neural predictivity of ANN-based models, along with the widely used linear regression neural prediction score. The paper reports following claims (1) ANN-based brain models can have similar neural prediction score but different sensitivity to adversarial probes, (2) Adversarial sensitivity have stronger correlation to internal model measurement such as sparseness and normalized variance, (3) Adversarially trained model share common perturbation subspace, while standard models do not, (4) adversarially trained models produces more semantically meaningful adversarial probes to the inputs. The analysis use fMRI dataset on Natural Scene Dataset, and a collection of CNN-based image models (VGG16, DenseNet, ResNet-50 Robust).

**Strengths:**

The paper is well-organized, with clear section organization and good figures for the readers to follow. The motivation is clear and well-defined, as is, how to address the problem that multiple ANN-based models have similar neural prediction score. The paper proposes to use adversarial probes as the approach to offer further discrimination between different models, and show that while models can have similar neural prediction score, they can have different sensitivity level to local perturbation produce by adversarial probes (Fig.2).

The experiments are clear and easy to understand. And the experimental results support the main claim that local perturbation geometry, as measured by the adversarial probes, can reveal additional information about the models further than the neural prediction score (sparseness and normalized variance in Fig 2C, transferability and alignment of the local geometry in Fig 3, and adversarial probes of robust models embeds more robust features.

However, despite these strengths, 3/4 main claims of the paper lacks novelty, and has been shown in previous works, which the paper did not explicitly mention. I will provide more details information about this limitation of the paper in the "Weakness" section

**Weaknesses:**

While the paper attempts to address a well-defined concern, that is, multiple ANN-based models have similar neural prediction score, the main contributions (3/4) have been seen in various previous works in both brain-based ANNs models and adversarial robustness literature, which the paper did not explicitly mention these previous contribution. Previous relevant works for each main claims are listed below:
1. Claim 1: Most contemporary ANN-based brain models are unexpectedly fragile, response predictions are highly sensitive to small, imperceptible perturbation, revealing unreliable local coding directions
	1. Adversarial examples are discovered in the seminal work of Szegedy et al, Intriguing properties of neural networks, which show that imperceptible perturbation to the inputs can lead large change in the model outputs, which is cited by 20000 times and developed into a well-defined research topic. Therefore, it's difficult to consider the paper findings that "ANN-based brain models are unexpectedly fragile, response predictions are highly sensitive to small, imperceptible perturbation" a surprising, novel or significant claims
	2. Related works:
		1. Szegedy, C., Zaremba, W., Sutskever, I., Bruna, J., Erhan, D., Goodfellow, I. and Fergus, R., 2013. Intriguing properties of neural networks. _arXiv preprint arXiv:1312.6199_.
2. Claim 2: Model's sensitivity to adversarial probes can better discriminate between candidate neural encoding models than prediction accuracy alone
	1. There's a significant body of works in the adversarial robust and brain-inspired model literature has shown that adversarially trained models have higher alignment with neural response. And in turn that incorporating brain-like features also improve adversarial robust of the model (Guo et al, Dapello et al, Dapello et al). In the machine learning literature, there are many papers reports that models with the same clean accuracy can have different robust accuracy (Goodfellow et al). Therefore, it's difficult to consider claim 2 as novel or significant.
	2. Related works:
		1. Guo, C., Lee, M., Leclerc, G., Dapello, J., Rao, Y., Madry, A. and Dicarlo, J., 2022, June. Adversarially trained neural representations are already as robust as biological neural representations. In _International Conference on Machine Learning_ (pp. 8072-8081). PMLR.
		2. Dapello, J., Kar, K., Schrimpf, M., Geary, R., Ferguson, M., Cox, D.D. and DiCarlo, J.J., 2022. Aligning model and macaque inferior temporal cortex representations improves model-to-human behavioral alignment and adversarial robustness. _BioRxiv_, pp.2022-07.
		3. Dapello, J., Marques, T., Schrimpf, M., Geiger, F., Cox, D. and DiCarlo, J.J., 2020. Simulating a primary visual cortex at the front of CNNs improves robustness to image perturbations. _Advances in Neural Information Processing Systems_, _33_, pp.13073-13087.
		4. Goodfellow, I.J., Shlens, J. and Szegedy, C., 2014. Explaining and harnessing adversarial examples. _arXiv preprint arXiv:1412.6572_.
3. Claim 3: Standard models rely on distinct local coding directions that do not transfer across model architectures
	1. The authors show that adversarial examples generated by adversarially trained models have higher transferability and alignment than the ones generated by standard models, which I found quite surprising given that there are many previous works in the ML community study the transferability of adversarial examples across different models, that is, adversarial examples created by a model (for example, ResNet-50), can be used to attack another model (for example, VGG-16) (Gu et al 2023). I would be great if the authors can comment on these discrepancy between their findings and previous literature.
	2. Related works:
		1. Xie, C., Zhang, Z., Zhou, Y., Bai, S., Wang, J., Ren, Z. and Yuille, A.L., 2019. Improving transferability of adversarial examples with input diversity. In _Proceedings of the IEEE/CVF conference on computer vision and pattern recognition_ (pp. 2730-2739).
		2. Tramèr, F., Papernot, N., Goodfellow, I., Boneh, D. and McDaniel, P., 2017. The space of transferable adversarial examples. _arXiv preprint arXiv:1704.03453_.
		3. Gu, J., Jia, X., de Jorge, P., Yu, W., Liu, X., Ma, A., Xun, Y., Hu, A., Khakzar, A., Li, Z. and Cao, X., 2023. A survey on transferability of adversarial examples across deep neural networks. _arXiv preprint arXiv:2310.17626_.
4. Claim 4: Adversarial probes from robustified models produce generalizable and semantically meaningful changes, suggesting that they capture local coding dimensions of visual systems
	1. In Figure 4, the authors show that adversarial probes created by robust model elicit more meaningful change to the inputs than the probes generated by standard models, and therefore can be used to design stimulus to modulate neural responses. About using artificial probes to modulate neural responses, Bashivan et al 2019 has shown that ANNs can be used to generate synthetic stimulus that trigger responses of specific neurons. About robust models encode more robust features, Ilyas et al 2019 has shown that both theoretically and empiricially, adversarially trained models learn more robust and interpretable features.
	2. Related works:
		1. Bashivan, P., Kar, K. and DiCarlo, J.J., 2019. Neural population control via deep image synthesis. _Science_, _364_(6439), p.eaav9436.
		2. Ilyas, A., Santurkar, S., Tsipras, D., Engstrom, L., Tran, B. and Madry, A., 2019. Adversarial examples are not bugs, they are features. _Advances in neural information processing systems_, _32_.

**Questions:**

1. Fig 2C. What is the definition of sparseness (is this the ratio of activated units) and normalized variance? What is the measurement reported here (0.18 and 0.06) (is it correlation)?
2. Fig 3B. What is the unit of the color bar? Is it L2 distance of the total change in neural response across M units?

---

> ### Author Response · Authors · 2025-11-15
> **Response 1 to Reviewer auvF**
>
> We thank the reviewer for the thoughtful and detailed assessment of our paper. It appears to us that the reviewer is satisfied with the quality, clarity, and rigor of the work, and is primarily raising questions about the novelty of our contributions which we address directly below.
>
> **Reg questions about novelty:** We appreciate the opportunity to tackle this questions head-on. In fact, the reviewer’s strong familiarity with both the adversarial-robustness and brain-modeling literatures is extremely helpful, because it allows us to precisely articulate how our contribution differs from prior work. We think part of the concern arises from conflating our methodological adaptation of adversarial probes with the conceptual and scientific novelty of applying them to the study of local representational geometry of brain-aligned encoding models of the brain, which has not been explored in any of the cited prior work (as we will show below).
>
> **Claim 1:** We of course agree that adversarial examples in classifiers were first identified by Szegedy et al., (which we have cited as well) and we do not claim novelty in the observation that classifier logits can be perturbed by small, imperceptible noise. As such in the context of classification models, adversarial fragility is not surprising.
>
> But novelty in our case is to show that prediction alone as a tests of brain models might be limited and misleading. Current brain-models might be predictive but they have *very different* local geometries.
>
> Our key insight (and novelty)  in this part is:
> A) the development of a new metric for brain models that goes beyond predictions.
> B) the use of adversarial perturbations to characterize the local representational geometry of brain-aligned models (what we formalize as the *local coding axis*.)
>
> **Change to the paper:** We are now clarifying the core contribution in this section more directly.
>
> In **Claim 2**, our aim is to demonstrate that brain-models with similarly high accuracy in fact have different local coding-axis geometry (we characterize using adversarial sensitivity). We also find that our metric better discriminates between candidate brain models.  To our knowledge, no other study has done this before!
>
> Here we are a little confused, because the literature cited in the review (Guo et al., Dapello et al., Goodfellow et al.), which we also cite and know intimately about, are in fact quite different from ours.
>
> Guo et al., characterize the robustness of single neurons in monkey IT cortex and compare them to representations in adversarially trained ANNs (with the brain, not AT-ResNets etc).
>
> Dapello et al., showing that adding a V1-like layer, makes standard models more robust. This is a result about an architectural inductive biases affecting classifier robustness.
>
> Dapello et al., (2022) demonstrate that regularizing models with macaque V4 and IT data improves robustness. Again, this concerns global classifier robustness and representational regularization with brain data.
>
> Goodfellow et al.,  is an ML paper characterizing adversarial perturbations in terms of classifier decision boundaries and behavioral outputs.
>
> While it is true that these studies explore relationships between robustness and the brain, this overlap is only superficial. The  goals, the analyses, the modeling spaces, brain data, and the core scientific inferences are fundamentally different. Clubbing our study together with these works therefore feels somewhat unfair.
>
> From a neural-coding perspective, our analysis focuses on the **local coding axis** of voxels in the brain. This is a fundamentally different object of study than classifier robustness or how using V1 frontends and/or neural regularization improves classifier robustness. None of the cited works study characterize the local neural geometry of neural encoding models.

---

> > ### Author Response · Authors · 2025-11-15
> > **Response 2 to Reviewer auvF**
> >
> > **Claim 3.**  We thank the reviewer for raising this point. Our result does not contradict the extensive ML literature on cross-model transferability of adversarial examples (e.g., Xie et al., Tramer et al., Gu et al.). In fact, it is fully compatible once we recognize that the questions, objectives, and spaces under comparison are fundamentally different.
> >
> > In this section, our goal is to probe the **local coding axis of brain-encoding models**: the voxel-level, first-order sensitivity of predicted neural responses. This can only be examined under a single-step, small-ε perturbation. Hence, we characterize the Jacobian of the encoding model in voxel-level fMRI prediction space, after the linear mapping from ANN features to neural responses, using single-step probes to isolate *local* representational geometry. I hope the reviewer agrees that *this* is the appropriate mathematical tool for asking: Do different brain models predict neural responses using the same local representational directions?
> >
> > We show that even equally predictive brain models have distinct local geometries: despite similar neural predictivity, their perturbation subspaces are distinct. We tried hard to find an overlap using three complementary metrics (which includes a highly sensitive energy-overlap measure) which converge to the same inference.
> >
> > With this framing, it becomes clear why our findings do not conflict with the ML transferability literature. Those studies evaluate transferability (in classifier decision space), typically using multi-step, transfer-optimized attacks (momentum, input diversity, iterative PGD variants) explicitly designed to *increase* cross-model transfer.
> >
> > Another way to think about this is that we *know* that brain-encoding models will also show transfer if perturbation norms are made large enough. They already make very similar predictions across many images (equally highly predictive). What we isolate and amplify in our work is that these same models can strongly disagree in the low-norm, first-order regime.
> >
> > We reveal a previously undescribed paradox:  Brain-models which agree globally (global representation) can disagree sharply locally (local coding axes).
> >
> > With this framing, it becomes clear why our findings do not conflict with the ML transferability literature. Those studies evaluate transferability (in classifier decision space), typically using multi-step, transfer-optimized attacks (momentum, input diversity, iterative PGD variants) explicitly designed to *increase* cross-model transfer.
> >
> > Another way to think about this is that we *know* that brain-encoding models will also show transfer if perturbation norms are made large enough. They already make very similar predictions across many images (equally highly predictive). What we isolate and amplify in our work is that these same models can strongly disagree in the low-norm, first-order regime.
> >
> > Revealing this paradox:  Brain-models which agree globally can disagree so sharply locally, is the novelty our study exposes.
> >
> > **Changes to the paper:** We are rewriting the motivation to clearly articulate why a single-step, local change is crucial to this analysis. We will also articulate exactly why other transfer methods are ill-suited for this task.

---

> ### Author Response · Authors · 2025-11-15
> **Response 3 to Reviewer auvF**
>
> **Claim 4:** We thank the reviewer for discussing this point and for bringing up the work of Bashivan et al., (which we’re also intimately familiar with). Note, however, that Bashivan et al. targeted  an intermediate visual cortex (V4) using extensive single-unit data from monkeys. In fact when people target later regions like V4, they always need to couple the encoder with a generator (like GAN/Diffusion models etc, see work from Ponce et al., for instance).  All studies in humans for this kind of thing either run high-throughput screening of naturalistic images or a generative prior like GANs/Diffusion model (Murty et al., Gu et al., Luo et al., among others).
>
> The reviewer raises a very good question though: *What does the optimization look like when (1) we only have limited fMRI data and (2) no access to a generator such as a GAN or diffusion model?*
> **So far we have not tested this; we are doing so now.**
>
> We also note that this comment from the reviewer implicitly aligns with one of our central claims: robust models are better candidates for brain modeling, especially in the context of causal, closed-loop perturbation experiments. This is precisely what our results reveal. The common zeitgeist in NeuroAI is that models with the highest neural predictivity are automatically the best models of the brain.  **Our findings show that this assumption is incomplete.** Even among models with matched neural predictivity, only robustified models yield stable, meaningful local coding directions. Neural predictivity alone does not guarantee biologically plausible local geometry, and robust models offer structural advantages that matter for causal neuroscience.
>
> This is the novelty of our work. To our knowledge our study is the most thorough characterization of the low-norm coding geometry of brain encoding models to date. This is why is it significant and meaningful.
>
> **Resp to Q1:** We thank the reviewer for the question.
> These are not correlations. We previously defined these metrics in Appendix A.1, and we happy to restate them here for clarity. *Sparseness* follows the classic Vinje & Gallant (2000) formulation widely used in neuroscience. It measures how selectively a prediction scores is concentrated across models, ranging from 0 (very similar prediction across models) to 1 (high discriminability between models).
>
> To compute *normalized variance*, we account for the fact that sensitivity is unbounded while predictivity is bounded between −1 and 1. Because these scales are not directly comparable, we first normalize each metric by dividing all values by its respective maximum value, and then compute the variance on these normalized quantities. This yields the normalized variance reported in Fig. 2C and ensures a fair comparison between metrics that live on different numerical ranges.
>
> **Change:** We will provide a short description to the main text linking the Appendix for the readers.
>
> **Resp to Q2:** Sorry about the confusion. The color is indeed the change in response. We will better clarify this in the figure description itself.
>
> Finally, we would be grateful for a renewed consideration of our study in light of the clarifications above. The reviewer clearly knows the relevant literature, and we hope that our detailed responses have articulated how the present work offers an important and valuable conceptual, theoretical, and methodological contribution to the NeuroAI community.

---

### Official Review · Reviewer_ukS1 · 2025-11-01

**Soundness:** 2
**Presentation:** 3
**Contribution:** 2
**Rating:** 4
**Confidence:** 4

**Summary:**

In this paper, the authors propose an image model’s robustness to noise as a discriminating factor among equally predictive neural encoding models in fMRI. They perform several analyses to better understand how robustness relates to encoding models, such as whether similarly predictive models share local adversarial directions (coding axes). This work adds to previous findings that show a tight coupling between neural encoding model robustness and robust activations in the brain.

**Strengths:**

Interestingly, the paper finds a number of neural models that show different interpretations of coding axes and provides an insightful data point on how underdetermined current state of the art neural encoding models are. The perturbation analysis is neat, it provides a clean way of measuring and comparing neural population sensitivity using a sensitivity subspace. In total, the paper explores an interesting question and demonstrates a novel analysis that should interest the computational neuroscience community.

**Weaknesses:**

The main weaknesses are that it analyzes only fit encoding networks with minimal links back to the brain, and it uses identical architectures for its “robust” models, which dilutes the impact of the results.

**Questions:**

One question I had while reading is to what extent the authors believe model sensitivity discrimination is distinct from neural predictivity. If the NSD stimulus set contained adversarial examples, such as small noise perturbations on top of images, then the current equally predictive models mentioned in the paper would likely separate in predictiveness. It could be that the most predictive encoding model is underdetermined by this dataset size rather than sensitivity being a distinct separation axis from predictivity.

The authors claim that robustness provides better discriminability than predictiveness alone. It is not clear that this is meaningful. These networks could also be cleanly separated by parameter count, training distribution, or any other meaningful difference in the model. The relation back to brain activity could be valuable, for example showing that robust models predict neural activity better in out of distribution cases, however the paper does not explore this. I find their claim 2 to be fairly weak because of this. Could the authors provide more evidence that sensitivity reflects a meaningful difference in neural activity rather than just discriminating between trained models.
While interesting, it is not clear that the analysis here provides meaningful interpretations for the brain. For example, the finding in Figure 3E that the coding axes of the robust networks are more similar to each other than those of the nonrobust networks is not necessarily an indication of neural encoding, and could instead indicate network similarity. These robust ResNet 50 models may simply be more similar to each other in training protocol, data, and architecture than the other models being compared.

The authors could strengthen their claim by addressing the following.

1. Can the authors relate their results to a testable prediction about neural activity. If the robust models more accurately capture true neural coding axes, they should have better generalization. One suggestion is to remove a distinct group of objects, for example images with faces, from training and evaluate on them during testing with robust and nonrobust models. Another is to examine scaling laws of encoding performance.
2. Could the authors provide results for a robustified image model that does not share the same architecture as the ones used here? There is a confound in the claim that robust models share more generalizable coding axes, since to my understanding they share the exact same architecture and differ only in adversarial objective strength.

---

> ### Author Response · Authors · 2025-11-25
> **Response 1 to Reviewer ukS1**
>
> We thank the reviewer for their careful reading of our work and for their very constructive feedback. We are encouraged that the reviewer recognizes the significance of the question, the novelty of the perturbation-based framework, and the value of comparing models based on their local coding axes (local sensitivity subspaces). The reviewer has raised two key concerns (the question about links back to the brain and the use of specific model architectures for robust models) and has been very constructive with their feedback. We are fully prepared to taking these questions head-on, and have run additional analyses to further improve the clarity and impact of our contributions.
>
> **Q1. Is sensitivity really distinct from predictivity?**  The reviewer’s thought experiment honestly made us chuckle (in a good way). It sparked a fun idea: imagine someone quietly uploads NSD images augmented with adversarial probes into the public dataset, and suddenly all current models fall flat and predictivity collapses. And the reviewer is exactly right in spirit: if the dataset contained adversarially perturbed images, some differences between models would start to show up in predictivity.
>
> But practically, adding adversarial probes would *not* collapse sensitivity into predictivity, for at least two reasons:
>
> 1. Adversarial probes do not generalize across models in the low-norm regime (as we show). A perturbation optimized for Model A will not necessarily move Model B in the same direction. Predictivity would then reflect idiosyncratic alignment with that model’s gradients, not the model-agnostic property of local coding.
>
> 2. More fundamentally it exposes issues with predictivity, which is a coarse, variance-dominated metric. It evaluates how well a model captures mean voxel/neural responses across thousands of natural images. It cannot not constrain a model’s local derivatives, which govern how responses change in the infinitesimal neighborhood of each image. Predictivity will be largely robust (think of it as adding some noise to the responses). Adding a handful of adversarial variants to the dataset would not resolve this underdetermination.
>
> So even with adversarial examples in the dataset, predictivity would remain an incomplete measure. Even if the dataset were very large (the predictive model is not underdetermined) predictivity would be incomplete as a measure.  It would sample a few additional off-manifold points, but it would still not tell us whether the models expose the local-coding axes (local-representational geometry) of models.
>
> To zoom out, our core point is that **global presentational metrics (like neural predictivity) must be complemented with other local-coding measures (like sensitivity and control-consensus scores, see next)**. The two capture *fundamentally* different (and complementary) aspects of what it means for a model to resemble the brain. Exposing this distinction is the theoretical and conceptual novelty of our study.
>
> **Q2a. Is adversarial sensitivity neuroscientifically meaningful?**
> We fully agree with the reviewer that simply separating models is not by itself a sufficient criterion. The separability should of course reflect something meaningful about neural encoding. We take this chance to clearly articulate two distinct neuroscientific links from our study that come from 2 distinct sections of the paper.
>
> Link 1 is from Claims 1 and 2 (which the reviewer has highlighted): In these sections we want to show that neural predictivity (a global representational measure) collapses different neural encoding models into the same score (which is now well-reported and replicated), even though these models implement very different local-coding geometries (robust to different degrees). We find that the field's current gold standards (neural predictivity, representational similarity, and encoding accuracy) are all just different flavors of the global agreement between model and brain. By their very design, they are completely insensitive to the fine-scale, first-order structure of a model’s representational geometry.
>
> This is the *neuroscientific* sense in which sensitivity provides “better discriminability”. To make the intuition more concrete, imagine we knew how robust the brain was and plot “the brain” alongside Figure 2. For the brain (at fMRI resolution), we hope the reviewer agrees that we expect the effect of low-norm perturbations to be effectively zero (we deliberately chose small-scale tests that are lower than the resolution-limit for humans). Yet our analysis shows that several models will predict a very large change in the response. This is a clear breakdown of our model which we expose. We want our brain-models to match all features of the brain. Brains are more robust than the leading prediction models. In this section we merely expose this issue. The core neuroscientific link in fact follows in Sections 3 and 4.

---

> ### Author Response · Authors · 2025-11-25
> **Response 2 to Reviewer ukS1**
>
> --contd Q2a--
>
> Link 2 (from Claims 3 and 4). In these sections, we turn it around and show the practical consequence of robustness. Adversarial probes do not generalize across models. For most standard models, the one-step local coding axes are highly idiosyncratic and fail to generalize. If all models were extracting the same underlying visual information (shared representational coding axes) from the images, their first-order perturbation directions should transfer. They do not.  Previously, this section was framed only as a transferability analysis. Based on the reviewer’s feedback, we have now strengthened the neuroscientific interpretation by introducing a brain-relevant metric:
>
> $$
> \text{Control-Consensus Score (CCS)} \=\ \frac{\text{mean predicted change in response in all other models}} {{\text{predicted change in response in source model}}}
> $$
>
> This metric directly quantifies how consistently a perturbation designed to change the response in one model is predicted to change responses in other models. A CCS close to 1 means the models share a common coding axis; near 0 means the perturbation direction is model-specific and does not generalize. We find that robustified models have consistently higher CCS scores than all baseline models.  This turns local-geometry alignment into a brain-relevant metric of consensus feature coding, which was missing before.
>
> Now our paper will have 2 metrics.
> Metric 1: Robustness, which is (admittedly) directly linked with robustifying models. Exposes that even highly predictive models are not robust.
> Metric 2: Control Consensus Score, which predicts the degree of change across models not directly related to robustness. Exposes that robustified models better transfer across models.
>
> Here is a table for what we find for all models.
>
> | Model               | Sensitivity | Control Consensus Score |
> |---------------------|-------------|--------------------------|
> | VGG16               | 2.9472      | 0.0341                   |
> | SqueezeNet v1.1     | 2.5658      | 0.0219                   |
> | MobileNet v2        | 2.4029      | 0.0268                   |
> | AlexNet             | 1.9280      | 0.0376                   |
> | Inception v3        | 1.7963      | 0.0240                   |
> | DenseNet 201        | 1.7947      | 0.0509                   |
> | CORnet\_RT          | 1.7827      | 0.0317                   |
> | ResNet-50           | 1.7799      | 0.0360                   |
> | CLIP-RN50           | 1.7502      | 0.0177                   |
> | Dino v2             | 1.7382      | 0.0276                   |
> | Nomic               | 1.6463      | 0.0274                   |
> | Dreamsim            | 1.6396      | 0.0429                   |
> | Google ViT          | 1.5630      | 0.0369                   |
> | Blip 2              | 0.7632      | 0.0472                   |
> | L2-RN50-robust-0.1  | 0.9876      | 0.1547                   |
> | L2-RN50-robust-1    | 0.4283      | 0.3530                   |
> | L2-RN50-robust-3    | 0.2793      | 0.5311                   |
> | L2-RN50-robust-5    | 0.1988      | 0.7372                   |
>
> The two metrics show an inverse association (Pearson R=.77, all models). As a model becomes more *sensitive*, its perturbations are less likely to transfer to other models. This association holds to a lower degree (R=-.40) even when we remove the adversarially trained models. Note however that the relation is better captured by a logistic regression (R2= 0.98) than a linear association.
>
> Changes to paper. We will now report both the adversarial sensitivity and the newly introduced Control Consensus Score (CCS). We will include clear statements for how each metric contributes to understanding robustness and generalization in brain-encoding models.
>
> **Q2b. Model separability using adversarial sensitivity relative to other factors like parameter count etc.**
> We of course agree that many factors can “separate” models in a purely descriptive sense (parameter count, training distribution, architecture family, etc.). However, our goal is not to separate models in an abstract way, but to separate them in a way that is meaningfully tied to brain responses.
>
> To clarify what we mean by “meaningful” separability, our metric is defined in neural-prediction space: adversarial sensitivity measures how the *predicted brain (fMRI) responses* changes under small, targeted perturbations. Parameter count, architecture labels, or dataset IDs do not live in this space at all. They are implementation details, not axes in brain space. Two models can have identical parameter counts and training datasets yet very different local coding directions, and conversely, two models with very different parameter counts could, in principle, induce similar local geometry. So comparing adversarial sensitivity to parameter count is not a fair comparison at all.

---

> ### Author Response · Authors · 2025-11-25
> **Response 3 to Reviewer ukS1**
>
> Crucially, we *do* relate sensitivity back to the brain. Section 4 of the paper demonstrates exactly how these local coding directions manifest in practice: robust models produce low-norm, semantically meaningful perturbations that generalize across models. This is especially suitable for causal closed manipulation of images to change neural responses. We show that adversarial probes generated from robustified models tend to change the predicted responses for all models and are also semantically meaningful. These features are not true of standard models which indicates that the adversarial probes are likely more aligned with brain-like features. Hence closed-loop control and semantically meaningful changes is where local-sensitivity becomes most meaningful.
>
> Finally, we would like to highlight that during the rebuttal period, new empirical evidence directly relevant to our claims was presented at the Society for Neuroscience conference (Wang et al., 2025, SFN abstract). This is based on an experiment performed on single monkeys from monkey V4 and IT cortex. Although this was not available at submission time, the findings strongly support our central claim about sensitivity being (A) very important to discriminate between models of the brain and (B) most relevant for neural control.
>
> https://www.abstractsonline.com/pp8/#!/21171/presentation/26662
>
> As the authors of the SFN abstract note, “standard models can achieve equivalent predictive scores yet rely on different—and sometimes spurious—features,” whereas “adversarially trained ResNets and ViTs consistently produced feature-accentuated images that modulated neurons in the intended parametric manner.”
>
> **This is exactly what we predicted in parts 1,2 and 4 of the paper (and especially with our new CCS)!** Taken together, these findings provide **independent, closed-loop (causal) evidence** that the distinctions revealed by our sensitivity metric correspond to meaningful separability and are distinct from factors like architecture etc.
>
> **Q1. Can the authors relate their results to a testable prediction about neural activity. If the robust models more accurately capture true neural coding axes, they should have better generalization.**
>
> We thank the reviewer again for this excellent question (see Link 2 above). We want to add that our analyses naturally lead to strong testable (and falsifiable) neuroscience predictions.
>
> 1. Prediction 1 : Small-scale perturbations that affect models should have no measurable effect on human fMRI responses. To be clear, this pattern is so striking and the effect is so large, that we feel it doesn’t even warrant running expensive fMRI studies (Link 1).
> 2. Prediction 2:  Adversarial probes generated from robustified models (and not baseline models) will reliably modulate neural responses in the human brain. We also predict that the degree of transfer to the brain will be related to the CCS score (Link 2).  We show that these probes produce meaningful changes to the stimulus (part 4). And there is some emerging evidence that this will likely hold even in other domains.
>
> We will now state these predictions explicitly in the discussions and make the link even more clear.
>
> Finally, we also want to explicitly acknowledge the reviewer’s suggestion regarding generalizability across stimulus categories and scaling-law–like comparisons. These are important directions for the field, but we note that both primarily concern global representational properties of models (e.g., distributional generalization, domain coverage, training scale). Because adversarial sensitivity isolates local coding geometry, we do not expect these manipulations which operate at the global representational level, to change the specific effects we report here. Nevertheless, we agree that they are worthwhile to explore in future work, and we appreciate the reviewer pointing them out.
>
> **Q2. Why not other model architectures.**
>
> We completely agree with the reviewer that limiting the analysis to robustified ResNet-50 variants introduced a (unintended) confound.
>
> We have now included 3 other model robustified model architectures (densenet, mobilenet and vgg16). All figures will be appended with these additional models. More importantly, all the core claims of the study still hold. (1) Robustified are predictive and stable to small-scale noise patterns (2) Adversarial probes from robustified models better generalize (high CCS score) to non-robust models (3) Adversarial probes from robustified models produce more semantically-related changes to the image than non-robustified models.

---

> > ### Author Response · Authors · 2025-11-25
> > **Response 4 to Reviewer ukS1**
> >
> > We thank the reviewer again for their constructive engagement with our work. Their comments helped us strengthen the paper substantially. In response, we (1) added a brain-relevant Control Consensus Score (CCS) to quantify consensus in local coding directions, (2) expanded the architectural diversity by including three additional robustified backbones (DenseNet, MobileNet, VGG16), (3) clarified the neuroscientific meaning of adversarial sensitivity and its complementarity to global predictivity metrics, (4) articulated explicit testable neural predictions, and (5) added new discussion sections on why existing transferability methods do not apply and why sensitivity exposes representational structure that global metrics cannot detect. Importantly, all of our central claims hold with these expanded analyses.  We respectfully invite the reviewer to reconsider their assessment of the study in light of these changes.

---

### Official Review · Reviewer_SPWp · 2025-11-03

**Soundness:** 2
**Presentation:** 3
**Contribution:** 3
**Rating:** 4
**Confidence:** 4

**Summary:**

In this submission, the authors present a quantitative analysis of the adversarial sensitivity of various encoding models of human vision, including adversarially trained encoding models. The authors highlight that robustness as a metric shows improved distinction across encoding models, making them a more suitable metric to rank the human-likeness of encoders. In addition, the authors also show that robustified ANNs produce relatively stronger transferrable perturbations highlighting that they pay attention to interpretable, object-centric image features. Their results suggest that adversarial sensitivity is correlated with similarity of encoding models to biological vision, and support the inclusion of adversarial robustness measurements to enhance our understanding of biological vision.

**Strengths:**

Strengths:
- The current submission tackles a very critical problem of developing novel metrics to better distinguish among encoding models of human vision. There are plenty of options to choose an encoding model, all of which show little inter-model variability in prediction accuracy. There is a clear need for complimentary methods to vet models in their ability to encode human-like visual representations.
- From a brief review of other works in this area, I agree with the authors that there is a scarcity of methods to better distinguish between encoding models of the brain apart from neural predictivity. Hence, I think the proposed direction addresses a pressing issue in the NeuroAI community.
- The paper is written really well; the figures are very well done to clearly illustrate a few ambiguous concepts like the distinctness of perturbation spaces through simple schematics (e.g. Fig. 3).
- I found Fig. 4 to be quite interesting in that there is a marked qualitative difference between adversarial perturbations generated for robustified models.

**Weaknesses:**

Weaknesses:
- Confound between adversarial training and adversarial sensitivity.
In their first contribution (Sec. 3-1), the authors compare the adversarial sensitivity of robustified and non-robustified ANNs to show that the former is less sensitive to adversarial perturbations. They use this result to highlight that non-robustified ANNs are brittle compared to robustified ones. This is not a novel finding; given the motivation of adversarial training is to reduce sensitivity to such perturbations.
- On the enhanced variance of noise sensitivity over neural predictivity.
A good metric is characterized by its ability to show individual differences between models. That said, I believe adding adversarial perturbations injects OOD noise into the images, hence increasing response instability across models. Hence, while I agree that adversarial sensitivity better separates models as a metric, I don't think this finding is also particularly novel / informative.
- Transferability of adversarial attacks.
The authors here claim that adversarial probes do not transfer across models, i.e., they all have distinct perturbation spaces. I believe this claim is not well supported given there have been many techniques in the adversarial machine learning community that have been designed to improve adversarial attack transferability (see [1]). Hence, I think the lack of transferability here is not ONLY a trait of the encoding models, but it is also caused by the specific adversarial optimization algorithm in question.

References:
1. Gu, J., Jia, X., de Jorge, P., Yu, W., Liu, X., Ma, A., ... & Torr, P. (2023). A survey on transferability of adversarial examples across deep neural networks. arXiv preprint arXiv:2310.17626.

**Questions:**

Additional notes:
Overall, I feel the presented results seem to all point to the improved adversarial robustness of robustified ANN models, and the improved distinguishability of encoding models based on adversarial sensitivity. I think these results are interesting, but I would love to see a quantitative way in which ranking models via adversarial sensitivity actually correlates with another metric of brain-likeness that the neuroscience community currently uses. At the current stage, I feel that the paper is highlighting various properties of adversarially robustified ANNs, but does not make a connection back to whether the proposed adversarial sensitivity metric, apart from better separating the models being compared, truly produces more human-like representations. If the authors could please comment on how a reader can be confident that ranking based on adversarial sensitivity implies more human-likeness, that would significantly strengthen the current submission.

**Details Of Ethics Concerns:**

NA.

---

> ### Author Response · Authors · 2025-11-16
> **Response 1 to Reviewer SPWp**
>
> We thank the reviewer for the thoughtful assessment and are very encouraged by their recognition of the core motivation and contributions of our work. We especially appreciate the reviewer’s acknowledgement of the current challenge in NeuroAI that ''many encoding models achieve similar neural predictivity, yet we lack principled tools to distinguish whether they truly capture human-like representations''. This is precisely what our work aims to address, and we are glad the reviewer sees the relevance and urgency of this direction.
>
> We would like to emphasize a second, equally important motivation behind our work: as predictive encoding models become increasingly powerful and are treated as ‘digital twins’, we need principled ways of characterizing what these models are actually representing, which is beyond responses predictions. Our framework directly addresses this need by revealing structure in the local coding geometry that standard predictivity do not access.
>
> We thank the reviewer for the opportunity to discuss these results further. We are confident that the clarifications and analyses we provide below directly address all of the reviewer’s concerns and further strengthen the contribution of the work.
>
> **1. Is there a tautology?** We fully understand the confusion and misunderstanding the reviewer is pointing to and in fact had the same concern. If adversarial training reduces sensitivity in classifier space, then why should it be surprising that robustified models also show lower sensitivity here? This would indeed be a valid concern *if* we were measuring adversarial sensitivity *at the classifier output level*. But that is not what we do. We would also like to acknowledge that the logic for why this tautological has been acknowledged in other ML mech-inter studies.
>
> The key point is that our “adversarial sensitivity” is computed entirely in neural-prediction space, after a separate linear encoding model is fit to fMRI responses. In this space, lower sensitivity is not guaranteed by adversarial classifier training.  The encoding models are not robustified by brain data (as in some studies like Guo et al., etc). Robustness in classifier space does not trivially imply robustness in activation space or the neural-prediction space.
>
> Importantly, it turns out that the mathematics developed for studying adversarial geometry in machine learning (first-order sensitivity, local gradients, Jacobian directions, and small-norm perturbations) is precisely the right formal machinery for studying the local coding geometry of the brain (in our case models since we have access to the gradients). We therefore co-opt adversarial perturbation analysis as a scientific tool because it is a principled way to interrogate how an encoding model’s predicted neural responses change in the immediate neighborhood of stimuli. This use of adversarial methods as a probe of neural coding axes is, to our knowledge, new, and is not addressed in prior ML robustness work.
>
> **Changes to the paper**. We will add a brief subsection titled *“Does robustifying a model trivially explain local coding directions?”* to clarify that adversarial training in classifier space does not determine the local coding directions of brain-encoding models. This addition will directly address the reviewer’s concern and make the conceptual distinction explicit for our readers.
>
> **2. Is adversarial noise just OOD noise?** This excellent point. In fact, this was *also* our concern at the outset: if adversarial sensitivity simply reflected arbitrary OOD noise injection, then the metric would be uninformative and potentially misleading. *This is exactly why we included shuffled-noise controls throughout the paper.* These controls use perturbations with identical magnitude and statistics but destroy the adversarial structure. These control noise patterns (equally OOD) produce minimal changes in predicted neural responses, whereas targeted perturbations produce large, model-specific effects.
>
> The reviewer says that this aspect is not novel or informative, we respectfully disagree. The key novelty and scientific value here is precisely that local coding directions (the first-order derivatives of the encoding model predictions wrt the image) are highly specific for each model. From a neuroscience perspective especially, this is highly informative. It reveals that models with matched neural predictivity encode distinct, structured, and meaningful local directions that the standard prediction metrics completely miss.
>
> **Change:** We will highlight the importance of the negative control noise patterns to guard against other explanations like the injection of OOD noise.

---

> > ### Author Response · Authors · 2025-11-16
> > **Response 2 to Reviewer SPWp**
> >
> > **3. Do models really do not transfer?** We thank the reviewer for raising this point. As we will show, our result does not contradict the extensive ML literature on transferability of adversarial attacks. Those papers study a different question, in a different space, using different attack regimes, with very different objectives.
> >
> > As a reminder, the focus of our goal is to expose local-coding axis of brain models. This specific task needs a a single-step, infinitesimal (low-norm) first-order tests. For this scientific question, we hope the reviewer agrees that the correct probe is the Jacobian, accessed via a single-step, infinitesimal (low-norm) first-order perturbation.
> >
> > Did we try hard enough? Given the neuroscientific context (focus on the *local* geometry of the brain-aligned space), and the resulting analysis constraints we rigorously tested the transferability of the local-coding directions. We examined both the directional alignment of the leading singular vectors (single most sensitive local direction) and the overlap of the full perturbation subspaces using 3 complementary metrics (cosine similarity, global subspace similarity and energy overlap). The results are consistent: equally predictive standard models occupy distinct local perturbation subspaces.
> >
> > Relation with other methods: It is true (and we're familiar with) that there is a large body of work on improving adversarial transferability across models. We have deliberately refrained from *blindly* applying those methods because their objectives are fundamentally misaligned with what we are measuring (the local coding axis). Concretely, they typically use multi-step, high-norm, or perceptibly altered perturbations that are explicitly optimized to *force* transfer across classifiers. That is a not a characterization of the natural local-coding axis. Perceptible changes would likely change the brain and we cannot test those. Furthermore we know that *models must transfer* in perceptible regimes simply because the models have high prediction accuracies on natural images and this would simple confirm global similarity.
> >
> > **Change:** We will add a short subsection that clarifies why existing transferability methods from the adversarial ML literature do not apply to our scientific question. This section will explain the distinction between high-norm, multi-step, transfer-optimized attacks used for classifiers and the low-norm, first-order probes required to study local coding directions in neural-prediction space.

---

> ### Author Response · Authors · 2025-11-16
> **Response 3 to Reviewer SpWp**
>
> **Q1. How does sensitivity relate with current quantitative metrics?**
>
> We thank the reviewer for this crucial question which gives us the chance to clearly articulate our results here.
>
> *First*, we would like to emphasize that any new metric should ideally not correlate with existing measures of brain similarity. The field's current gold standards (neural predictivity, representational similarity, and encoding accuracy) are all just different flavors of the global agreement between model and brain. By their very design, they are completely insensitive to the fine-scale, first-order structure of a model’s representational geometry. This is the exact the thing we wanted to expose.
>
> *Second*, we want to be careful and acknowledge that no single metric is ever going to be sufficient to capture the full complexity of brain-likeness. A complete evaluation will require a *suite* of complementary metrics. The problem is, our current toolkit is almost entirely one-note. For now what we are saying is that prediction may not be sufficient and sensitivity provides a complementary metric. By the same logic, having sensitivity without predictivity is not useful either. While models look like they’re plateauing on the old measures (prediction), they very much can *still* be differentiated using sensitivity.
>
> The reviewer asks how a reader can be confident that our sensitivity metric is more than just a new way to separate models. We are cautiously optimistic based on the fact that our metric is the *first* to quantify this local alignment, and this robustness is now being independently and quantitatively linked by other groups to key measures of "human-likeness".
>
> Feather et al., Nature Neuroscience, show that robustifying  models improves recognizability of both visiual and auditory model metamers (see Fig 4 and 5 of their paper).
>
> Skill et al., (2025), CCN proceedings, show that robustifying models improves model prediction accuracies when using controversial stimuli.
>
> Finally Doshi et al., (2025) arXiv show using anagrams that robustifying models improves shape scores.
>
> These results strongly suggests that robustness is a critical, missing ingredient for brain-like computation. Our work provides the first theoretical and quantitative link explaining *why* these behavioral and neural observations are connected.
>
> *Finally*, we understand the reviewer’s concern about the practical relevance of this sensitivity metric. This is precisely why we performed the 'fixed-effect, variable-budget' analysis in Section 4. We wanted to demonstrate a concrete, practical application for these local coding axes. One critical place this metric matters is in generating targeted neural stimuli *without* relying on a powerful generative prior (like a GAN or diffusion model). Our method allows us to synthesize interpretable, hypothesis-driven stimuli in a constrained, low-norm regime.
>
> In summary, our analyses expose a central paradox: **Global agreement is not local alignment.** High representational similarity and neural predictivity (the field's current gold standard) *do not imply* that a model's local coding directions match the brain's. This disconnect is precisely what our targeted analyses reveal, and it represents a scientifically meaningful insight that existing methods have been blind to.
>
> **Changes:** We are making three changes. First we will explain how prediction and sensitivity measure different things and that we need both. Second, we will discuss and link our results with independent analyses from other groups. Finally we are running some additional generative model experiments.
>
> Thank you again for your thoughtful feedback. We have addressed all concerns directly and comprehensively, and we will follow up with the corresponding analyses and revisions to the paper. We are fully prepared to start the discussions now, and we respectfully request a renewed consideration of our work in light of these clarifications and additions.

---

### Author Response · Authors · 2025-12-03
**Global comment 1**

We sincerely thank all reviewers for their detailed comments and thoughtful evaluations which have made the paper stronger and clearer. We have addressed all comments, and the manuscript now provides clearer framing and stronger justification. Below, we summarize the key updates to our paper:

**1. Adding a Control-Consensus Score**

We acknowledge that the adversarial sensitivity, while useful for discriminating between equally performing neural encoding models, may not appear be a sufficient enough criterion for identifying the most brain-like models. In the first two claims of the paper, we use adversarial sensitivity only to illustrate the point that *global* alignment metrics (predictivity, representational similarity) do not imply *local* alignment (the first-order principle directions of the models).

We use this fact to motivate Claims 3 and 4, which demonstrate the consequences of  sensitivity: adversarial probes do not transfer across models, suggesting that the models are not actually extracting and encoding the same visual information. We have strengthened this claim by introducing a **control-consensus score (CCS)**, which quantitatively measures how consistently a perturbation designed to change the response in one model is predicted to change responses in other models. We find that adversarially trained models have the highest **CCS**, turning local-geometry alignment into a brain-relevant metric of consensus feature coding.

Our paper now contains two metrics: 1) adversarial sensitivity, exposing that even highly predictive models are not robust, and 2) a control-consensus score, exposing that robustified models better transfer across models. We find that the adversarial perturbations from robustified models tend to produce more semantically structured perturbations, which we hypothesize align closer with the principle local coding directions of the actual brain.

**2. Clarification regarding key claims / novelty**

We understand the reviewers’ intuition that neural encoding models may inherit adversarial sensitivity from their pretrained architecture backbones (Claim 1). While we agree that this result may feel intuitive, our goal here is to simply point out that *prediction alone is not a sufficient test of brain-likeness*. We use this initial claim to motivate the central claims in the paper, asking how we can characterize the local geometry. Importantly, models with low sensitivity scores also exhibit several other interesting properties (they achieve higher control-consensus scores, converge on shared representational geometries, and exhibit semantically structured principle directions). We find that these are not solely attributes of low sensitivity. Our work suggests that these are properties of adversarial training, contributing to a growing literature connecting adversarially trained models and “human-likeness”. Finally, we note that this question is separate from the general goal of finding “universal” attacks in the machine learning literature. These studies typically use multi-step or perceptibly altered perturbations that are explicitly optimized to force transfer across classifiers. In our study, we instead utilize low-norm, first-order probes to study the local coding directions in neural prediction space. We have clarified these points in the manuscript and believe that this clarification has significantly strengthened our presentation of the core conceptual claims.

**3. Expansion of robustified models**

We agree that only including robustified ResNet-50 models introduces an unintended confound, as these models share the exact same architecture and differ only in adversarial objective strength. We ran experiments using three other model robustified model architectures (DenseNet, MobileNet and VGG16). Importantly, we found strong alignment between the perturbation subspaces of all robustified models, regardless of the architecture. We note that even a standard ResNet50 and adversarially trained ResNet50 occupy very different perturbation subspaces, whereas two adversarially trained models with different architectures (e.g. robustified ResNet-50 and robustified VGG) share significant overlap. We have updated this figure in the paper and will update all other figures (such as general predictivity and sensitivity scores for the new models) accordingly.

---

> ### Author Response · Authors · 2025-12-03
> **Global comment 2**
>
> **4. Clarification in Discussion**
>
> We have substantially extended the discussion by providing examples of how adversarial probes can be used to identify new neuroscientific discoveries (such as identifying the attributes that drive neural responses most strongly in a given brain region and reverse-engineering stimuli without naturalistic generative priors). In addition, we have discussed and linked our study with work from other groups. Finally, we we provided testable and falsifiable predictions related to our work, such as the notion that adversarial probes generated from robustified models (and not baseline models) should reliably modulate neural responses in the human brain).
>
> ---
>
> The new version of the paper now presents a stronger and more clear account of global versus local alignment in neural encoding models.  We believe these revisions fully address the reviewers’ concerns, and we are very grateful for the reviewers’ thoughtful comments which have strengthened the work substantially.

---

### Meta-Review · Area_Chair_8Lts · 2026-01-04

**Summary:**

This paper studies local representational geometry of ANN-based brain encoding models using small, targeted adversarial perturbations, arguing that robustness-based measures reveal differences not captured by neural predictivity alone and favor robustified models. Reviews agree the analysis is careful and the question is interesting, but overall enthusiasm was limited, with concerns about novelty, interpretation, and how strongly the proposed metrics connect back to brain-likeness.

**Reviewer Concerns:**

Issues partially addressed in the rebuttal:
* concern that initial findings about the fragility of standard models are expected, given known adversarial vulnerability -> authors clarify this is a diagnostic step motivating later claims
* questions about whether adversarial sensitivity is merely injecting OOD noise -> authors add shuffled-noise controls and clarify the low-norm, first-order regime
* confound that robustified models initially shared architecture -> authors add results for additional robust architectures
* requests for clearer discussion and testable predictions -> authors expand discussion and articulate falsifiable predictions

Issues that remain concerns:
* novelty: multiple reviewers argue that most core claims overlap with prior work in adversarial robustness and NeuroAI, and that 3/4 claims are incremental rather than clearly new
* link to brain-likeness: reviewers question whether better model separability via sensitivity or CCS actually implies more human-like neural representations, versus just distinguishing models
* transferability interpretation: reviewers note that lack of transfer depends on attack regime, and that conclusions may reflect probe choice rather than fundamental representational differences
* conceptual framing: some reviewers remain unconvinced that sensitivity/CCS go beyond intuitive consequences of robustness and linear readouts
* significance: several reviewers state results are interesting but not clearly strong enough for acceptance

As AC, I find this to be a solid, carefully executed study, but one that falls short in terms of impact. The rebuttal is thorough and improves clarity, especially by adding controls and expanding model coverage. However, the central concern remains that much of the story aligns with existing robustness results, and the case that these metrics provide a qualitatively new or decisive criterion for brain-likeness is not fully convincing. No reviewer strongly advocates for acceptance, and all scores are below threshold.

**Reviewer Scores:**

Reviewer SPWp: 4, likely unchanged
Reviewer ukS1: 4, possibly slightly higher but still borderline
Reviewer Qijk: 4, unchanged
Reviewer auvF: 2, unchanged

---

### Decision · Program_Chairs · 2026-01-26

Reject